# OFFLINE SAFE POLICY OPTIMIZATION FROM HUMAN FEEDBACK

## ABSTRACT

Offline preference-based reinforcement learning (PbRL) learns rewards and policies aligned with human preferences without the need for extensive reward engineering and direct interaction with human annotators. However, ensuring safety remains a critical challenge across many domains and tasks. Previous works on safe RL from human feedback (RLHF) first learn reward and cost models from offline data, and then use constrained RL to optimize a safe policy. However, inaccuracies in the reward and cost learning can impair performance when used with constrained RL methods. To address these challenges, (a) we introduce a framework that learns a policy based on pairwise preferences regarding the agent's behavior in terms of rewards, as well as binary labels indicating the safety of trajectory segments, without access to ground-truth rewards or costs; (b) we combine the preference learning module with safety alignment in a constrained optimization problem. This optimization problem is solved using a Lagrangian method that directly learns reward maximizing safe policy without explicitly learning reward and cost models, avoiding the need for constrained RL; (c) to evaluate our approach, we construct new datasets with synthetic human feedback, built upon a well-established offline safe RL benchmark. Empirically, our method successfully learns safe policies with high rewards, outperforming baselines with ground-truth reward and cost, as well as state-of-the-art RLHF approaches.

## 1 INTRODUCTION

To align the intelligent agents with human values, preference-based reinforcement learning (PbRL) (Wirth et al., 2017) (also known as Reinforcement Learning from Human Feedback (RLHF)) has emerged as a popular learning paradigm by training the agent's policy from human pairwise preference over agent's behavior while no reward engineering is required. Offline PbRL (Shin et al., 2021) addresses the problem of feedback efficiency by avoiding online interactions with human annotators. It recently has shown great success when applied to policy learning in control tasks (Christiano et al., 2017; Lee et al., 2021; Park et al., 2022; Hu et al., 2024; Hejna et al., 2024) and finetuning large language models (LLMs) (Rafailov et al., 2024; Zhao et al., 2023; Ethayarajh et al., 2024). While it demonstrates the ability to learn rewards and policy that are consistent with human preferences, guaranteeing safety remains a significant challenge for PbRL. For instance, a robotic agent should avoid collisions with the environment and human co-workers during a control task. For finetuning LLMs, the agent should always generate harmless content without violating social norms and morals. In this paper, we aim to learn a policy that is

- consistent with human pairwise preferences, and
- aligned with human safety considerations.

Recently, safe RLHF (Dai et al., 2024) provides an approach for finetuning LLMs to generate helpful, harmless responses. It assumes access to *pairwise* human preferences for *both* rewards and costs as well as safety labels for each response. The proposed method learns reward and cost models from human feedback, based on which a policy is optimized using constrained RL. In this paper, we focus on the continuous control tasks which present additional challenges compared to safe RLHF for LLMs. LLMs finetuning is viewed as a contextual bandit problem, while continuous control tasks are usually modeled as a sequential decision-making process using RL. Additionally, safe RLHF

method follows the conventional two-phase PbRL learning paradigm. In the first phase, reward and cost models are learned from human feedback. In the second phase, constrained RL is applied to optimize a policy based on the learned reward and cost. However, inaccuracies during the reward and cost learning phase can undermine policy performance, and rollout sampling or bootstrapping in the constrained RL phase may introduce optimization challenges.

In this paper, we introduce the framework of *Offline Safe Policy Optimization from Human Feedback (POHF)*, where a policy is learned using offline datasets that include two types of human feedback: (a) *pairwise* preferences of agents' behavior regarding rewards, and (b) *binary* labels indicating whether the trajectory segment is safe or not. Notably, pairwise preference of agent's behavior regarding cost is not required as in prior work (Dai et al., 2024) since it is relatively scarce and expensive to collect in practice (Ethayarajh et al., 2024). To address the aforementioned challenges, we first present a novel method for learning a policy that generates safe behavior based solely on safety labels. Next, we transform the safety alignment objective into an optimization *constraint* by demonstrating that it implicitly defines a feasible set of policies. This safety alignment module is then integrated into the preference alignment module and the Lagrangian method is employed to learn a policy directly using the offline dataset. As a result, we derive a fully supervised learning objective, eliminating the need for additional reward and cost learning and constrained RL.

To evaluate our approach, we constructed a new dataset based on the well-established offline safe RL benchmark, DSRL (Liu et al., 2023a). We synthesized human feedback using the ground truth rewards and costs provided in the original offline dataset as they are not accessible during training in our setting. The dataset includes 29 continuous control tasks across two prevailing domains, providing a robust platform for testing offline safe policy optimization from human feedback and facilitating further study. We compare our method against baselines from offline safe RL (Liu et al., 2023a; Xu et al., 2022; Liu et al., 2023b), which use ground truth reward and cost, as well as offline RLHF approaches, including a variant of safe RLHF. The results suggest that, compared to the baselines, our method effectively learns a policy that achieves high rewards while adhering to constraints implicitly encoded in the human feedback.

The contributions of this paper are threefold. First, we introduce the Offline Safe POHF framework for control tasks, where a policy is learned using the offline dataset including pairwise agent behavior preferences related to reward, and binary safety labels for each trajectory segment. Second, we propose a practical solution by integrating preference and safety alignment modules to define a constrained optimization objective, and the policy is optimized using the Lagrangian method. This approach removes the need for reward and cost learning, as well as an additional constrained RL phase. Third, we extensively evaluate our approach using newly synthesized datasets, demonstrating the effectiveness of the proposed method.

## 2 RELATED WORK

**Preference-based Reinforcement Learning**   To avoid reward engineering which requires expert knowledge and may suffer from negative effects of reward misspecification (Pan et al., 2022), Preference-based Reinforcement Learning (PbRL) (also known as Reinforcement Learning from Human Feedback (RLHF)) provides a promising paradigm to learn a policy from human feedback (Wirth et al., 2017). There has been significant advances recently, both for control tasks (Christiano et al., 2017; Lee et al., 2021; Park et al., 2022; Liu et al., 2022; Shin et al., 2021; Hejna & Sadigh, 2024; Kang et al., 2023; Hejna et al., 2024) and LLM finetuning (Ziegler et al., 2019; Stiennon et al., 2020; Ouyang et al., 2022; Bai et al., 2022; Zhao et al., 2023; Rafailov et al., 2024; Ethayarajh et al., 2024). Besides preference alignment, researchers also start to focus on how to ensure safety for policy learning. Safe RLHF (Dai et al., 2024) is proposed to finetune an LLM that generates helpful and harmless responses. This approach is specifically designed for LLMs, meaning it applies to contextual bandits settings, rather than RL, which is a multi-step sequential decision-making setting. Safe RLHF needs to explicitly learn reward and cost models and perform constrained RL using such learned models. While effective, the inaccuracies in reward and cost model learning may adversely impact the policy learning using constrained RL. In this paper, we introduce a framework for Offline Safe Policy Optimization from Human Feedback (POHF) for continuous control tasks, and aim to learn a safe policy directly with pairwise preference and binary safety labels.

**Offline Safe Reinforcement Learning**   Offline safe RL provides a practical framework for learning safe policies using pre-collected datasets (Liu et al., 2023a). Previous work has addressed this problem through various approaches such as sequential modeling (Liu et al., 2023b), distribution correction estimation (Lee et al., 2022), Q-learning (Xu et al., 2022), and feasible region identification (Zheng et al., 2024). These offline datasets typically consist of agent rollouts with ground truth rewards and costs for each timestep, often designed by experts to ensure the data quality for policy learning. However, in many complex real-world domains and tasks, it is difficult to manually design reward and cost functions that accurately reflect human values. Therefore, with the advances of PbRL, we propose to learn policies from pre-collected human feedback, replacing ground truth rewards and costs with human pairwise preferences regarding agent behavior for rewards and binary labels indicating whether the behavior is safe or not.

## 3   PROBLEM DEFINITION

We model our problem as a Markov Decision Process (MDP), denoted as a tuple $\mathcal{M} = (\mathcal{S}, \mathcal{A}, \mathcal{P}, r, \rho_0, \gamma)$. $\mathcal{S}$ is the state space, $\mathcal{A}$ is the action space. $\mathcal{P}(s'|s, a)$ denotes the transition dynamics. $r : \mathcal{S} \times \mathcal{A} \to \mathbb{R}$ is the reward function. $\rho_0$ is the distribution of starting state and $\gamma$ is the discount factor. Safe RL is usually modeled as constrained MDP, $\mathcal{M} \cup \mathcal{C}$, where $\mathcal{C} = \{(c_i, b_i)\}_{i=0}^m$. $c_i$ is the cost function and $b_i$ is the cost threshold. The discounted cumulative reward with respect to a policy $\pi$ is defined as $V_\pi^r(s) = \mathbb{E}_\pi[\sum_{t=0}^\infty \gamma^t r(s_t, a_t)|s_0 = s]$. The discounted cumulative cost with respect to a policy $\pi$ is $V_\pi^{c_i}(s) = \mathbb{E}_\pi[\sum_{t=0}^\infty \gamma^t c_i(s_t, a_t)|s_0 = s]$. The objective function of such a constrained RL problem is:

$$\max \mathbb{E}_{s \sim \rho_0}[V_\pi^r(s)], \text{ s.t., } \mathbb{E}_{s \sim \rho_0}[V_\pi^{c_i}(s)] \le b_i, \forall i \tag{1}$$

In this paper, we consider learning a policy using an offline dataset. We assume no access to ground truth reward and cost. Instead, pairwise preference between trajectory segments regarding reward, and binary safety labels for each segment are provided. Since there are two different types of human feedback in the offline dataset unlike the preference-only approach typical in offline PbRL, we refer to our problem framework as *Offline Safe Policy Optimization from Human Feedback (POHF)*[1], as shown in Figure 1. Typically, we are given a dataset $D = \{(\sigma^+, y^+, \sigma^-, y^-)\}$, where $\sigma = (s_1, a_1, s_2, a_2, \cdots, s_k, a_k)$ is a $k$-length trajectory segment. The human preference between two segments is expressed as $\sigma^+ \succ \sigma^-$, with $+$ indicating the preferred segment and $-$ the unpreferred segment. Each segment is also as-

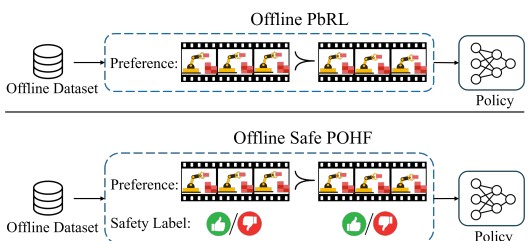

Figure 1: Offline Safe POHF versus Offline PbRL for control tasks. Besides pairwise preference between agent's trajectory segments, the dataset of offline safe RLHF additionally includes binary safety labels of each segment, which is used to align the policy with implicit safety constraints.

signed a binary safety label $y \in \{-1, +1\}$ where $y^+$ is the safety label of $\sigma^+$ and $y^-$ is the safety label of $\sigma^-$, with $-1$ denoting unsafe and $+1$ indicating safe. Our goal is to learn a policy that maximizes rewards while ensuring safety, using the segment preferences and safety labels. To address the challenges, we first decompose the problem into two modules, preference alignment and safety alignment, and then integrate them into a constrained optimization problem. In the following sections, we will discuss the each module in detail individually and propose an approach to merge them into a unified objective function.

## 4   PRELIMINARIES

**Contrastive Preference Learning (CPL) for Preference Alignment**   To learn a policy that aligns with human pairwise preferences, we consider a preference dataset $D_{\text{pref}} = \{(\sigma^+, \sigma^-)\}$. The con-

---

[1]The main difference from Safe RLHF (Dai et al., 2024) is that the human preference is for reward only. The preference of agent's behavior related to cost is not required.

ventional PbRL learning paradigm offers a class of methods that typically involve two phases (Christiano et al., 2017). In the first phase, it assumes the human preference model is distributed according to cumulative rewards and a reward model is learned by optimizing the negative log-likelihood of human preferences. In the second phase, a policy is trained by using the learned reward model. However, there exist two main challenges of applying this conventional PbRL paradigm. First, the assumption that human preferences are reward-based has been criticized as incompetent at capturing true human preference (Knox et al., 2024). Second, the RL training in the second phase would suffer from substantial computational difficulties (Hejna et al., 2024; Rafailov et al., 2024).

To address the issues, Contrastive Preference Learning (CPL) (Hejna et al., 2024) has been proposed as a method to learn a policy directly without the need for reward learning and RL. In CPL, human preference is modeled using regret instead of reward (Knox et al., 2024). By leveraging the equivalence of negated regret and the discounted sum of optimal advantages, the regret-based preference model is given by:

$$P[\sigma^+ \succ \sigma^-] = \frac{\exp \sum_{\sigma^+} \gamma^t A_r^*(s_t^+, a_t^+)}{\exp \sum_{\sigma^+} \gamma^t A_r^*(s_t^+, a_t^+) + \exp \sum_{\sigma^-} \gamma^t A_r^*(s_t^-, a_t^-)} \quad (2)$$

where $A_r^*(s_t, a_t)$ is the optimal advantage function of a single timestep $(s_t, a_t)$ with respect to a reward model $r$; shorthand "+" and "-" index the states and actions of segments $\sigma^+$ and $\sigma^-$. According to the principle of maximum entropy (Ziebart, 2010; Hejna et al., 2024), the optimal advantage function $A_r^*(s, a)$ for a Kullback–Leibler divergence (KL)-regularized RL problem can be expressed in terms of the optimal policy $\pi^*$ as:

$$A_r^*(s, a) = \alpha \log \frac{\pi^*(a|s)}{\pi_{\text{ref}}(a|s)} \quad (3)$$

where $\pi_{\text{ref}}$ is a reference policy used to regularize $\pi^*$, and $\alpha$ is a temperature parameter that determines the extent to which the reference policy $\pi_{\text{ref}}$ influences $\pi^*$. Consequently, the loss function for learning a parameterized policy $\pi_\theta$ (i.e., an approximation of $\pi^*$) is formulated by optimizing the negative log-likelihood of human preferences:

$$L_{\text{CPL-KL}}(\pi_\theta, D_{\text{pref}}) = \mathbb{E}_{(\sigma^+, \sigma^-) \sim D_{\text{pref}}} \left[ -\log P_{\pi_\theta}[\sigma^+ \succ \sigma^-] \right] \quad (4)$$

With Equation 3, the loss function $L_{\text{CPL-KL}}(\pi_\theta, D_{\text{pref}})$ presents a closed-form formulation for directly learning a policy that aligns with human preferences.

**Prospect Theory and Human-Aware Losses**   Prospect theory is a behavioral economics framework that explains how individuals evaluate gains and losses in uncertain events, often in an asymmetric manner. There are three key principles when modeling human decision-making through the lens of prospect theory, *a)* the use of a reference point to determine relative gains or losses, *b)* concavity in relative gains (i.e., diminishing sensitivity as they move farther from the reference point); and *c)* loss aversion, meaning individuals are more sensitive to losses compared to gains. Building on prospect theory, Ethayarajh et al. (2024) introduced a family of human-aware losses (HALOs) to to better understand the mechanism of RLHF for finetuning LLM. A function $f$ is a HALO such that:

$$f(\pi_\phi, \pi_{\text{ref}}^{\text{LLM}}) = \mathbb{E}_{x, y \sim D_{\text{LLM}}}[a_{x,y} v(r_{\pi_\phi}(x, y) - \mathbb{E}_Q[r_{\pi_\phi}(x, y')])] + C_{D_{\text{LLM}}} \quad (5)$$

Here, $\pi_\phi : \mathcal{X} \to \mathcal{P}(\mathcal{Y})$ is the parameterized LLM model to be aligned, where $x \in \mathcal{X}$ is an input to a LLM, $y \in \mathcal{Y}$ is a response generated by the model. $\pi_{\text{ref}}^{\text{LLM}}$ is the reference model for regularizing $\pi_\phi$. The sign $a_{x,y} \in \{+1, -1\}$ indicates whether the outcome is a gain or a loss. The implied reward $r_{\pi_\phi}$ is defined as $r_{\pi_\phi}(x, y) = l(y) \log[\pi_\phi(y|x) / \pi_{\text{ref}}^{\text{LLM}}(y|x)]$, where $l : \mathcal{Y} \to \mathbb{R}^+$ is a normalizing factor. $Q(Y'|x)$ is a reference point distribution over $\mathcal{Y}$, and $v : \mathbb{R} \to \mathbb{R}$ is a non-decreasing function that is concave over $(0, \infty)$. $D_{\text{LLM}}$ is the feedback data and $C_{D_{\text{LLM}}} \in \mathbb{R}$ is a data-specific constant. As the LLM is finetuned by optimizing the loss function $f$, the model is expected to assign higher rewards to desirable responses through $r_{\pi_\phi}$ (i.e., $\pi_\phi$ is more likely to generate desirable responses) and lower rewards to undesirable ones. Empirically, HALOs approaches either match or outperform non-HALO methods across various scales of LLMs (Ethayarajh et al., 2024).

## 5   OFFLINE SAFE POLICY OPTIMIZATION FROM HUMAN FEEDBACK

In the Offline Safe POHF setting, we assume access to additional safety annotations $y \in \{-1, +1\}$ for each trajectory segment. Setting aside pairwise preferences, we can frame safety alignment as

policy optimization based solely on binary feedback. Ethayarajh et al. (2024) recently introduced a novel approach (i.e., KTO) for finetuning LLMs using binary safety feedback. However, KTO is tailored specifically for contextual bandit settings. In section 5.1, we show how to extend the ideas behind KTO, such as prospect theory (Tversky & Kahneman, 1992) and human aware losses (HALOs), to our sequential decision making setting. Additionally, CPL (Hejna et al., 2024) is developed for learning from pairwise preference feedback, it does not take into account the safety aspect. We cannot optimize safety and reward preferences independently. Therefore, one of our key contributions is the principled integration of safety modeling from prospect theory perspective with preference optimization in a sequential safe RL setting, which will be discussed in detail in section 5.2.

## 5.1 SAFETY ALIGNMENT WITH BINARY SAFETY LABELS

Inspired by prospect theory (Tversky & Kahneman, 1992) and human-aware losses (HALOs) (Ethayarajh et al., 2024) which have been applied to align LLM with human values using binary signals of desirability, we derive the objective function for safety alignment in the context of continuous control tasks.

### 5.1.1 SAFETY ALIGNMENT FOR CONTINUOUS CONTROL TASKS

Although HALOs are introduced for the LLM setting (i.e., the contextual bandit framework), they offer a novel perspective for understanding safe RLHF. For continuous control tasks which involve multi-step sequential decision-making, we derive a loss function specifically for safety alignment. We begin by defining the utility function of a trajectory segment $\sigma$ as:

$$u(\sigma) \triangleq \psi_\pi(\sigma) - z_{\text{ref}} \tag{6}$$

where $\psi_\pi(\sigma)$ produces a scalar score of $\sigma$ based on policy $\pi$, and $z_{\text{ref}}$ serves as a reference point that determines the relative gain or loss when evaluating the outcome of $\sigma$ against all possible trajectory segments; score function $\psi_\pi(\sigma)$ is analogous to $r$ in Equation 5. Inspired by the definition of HALOs, we can express $\psi_{\pi_\theta}(\sigma)$ with a parameterized policy $\pi_\theta$ [2] as follows:

$$\psi_{\pi_\theta}(\sigma) = \sum_{t=0}^{T} \gamma^t \beta \log \frac{\pi_\theta(a_t|s_t)}{\pi_{\text{ref}}(a_t|s_t)} \tag{7}$$

It is the cumulative logarithm of $\pi_\theta$ along the trajectory segment, regularized by the reference policy $\pi_{\text{ref}}$, with the hyperparameter $\beta$ governing the degree of influence $\pi_{\text{ref}}$ has on $\pi_\theta$. Accordingly, we define $z_{\text{ref}}$ as the expected score of all segments $\sigma$ that humans have encountered under policy $\pi_\theta$ using offline data, which serves as a biased estimate of the ground truth reference point, i.e., $\hat{z}_{\text{ref}} = \mathbb{E}_\sigma \left[ \sum_{t=0}^{T} \gamma^t \beta \log \frac{\pi_\theta(a_t|s_t)}{\pi_{\text{ref}}(a_t|s_t)} \right]$. To ensure stable training, we do not backpropagate through $\hat{z}_{\text{ref}}$, it exists solely to regulate the loss saturation (Ethayarajh et al., 2024).

To develop a feasible objective function, we adopt the practices outlined in Ethayarajh et al. (2024) to facilitate optimization. We use `sigmoid` function as $v$ in HALO, as it aligns with the principle of prospect theory by being concave in gains and convex in losses. Additionally, we introduce two weight values, $\lambda_s$ and $\lambda_u$, for safe and unsafe segments respectively. These weights indicate the importance of $\sigma$ during policy training and reflect the concept of loss aversion in prospect theory (Tversky & Kahneman, 1992). Therefore, the loss function for safety alignment is written as:

$$L_{\text{safety}}(\pi_\theta, D) = \lambda_s \mathbb{E}_{\sigma \sim D_{\text{safe}}} \left[ 1 - \texttt{sigmoid}\left(u(\sigma)\right) \right] + \lambda_u \mathbb{E}_{\sigma \sim D_{\text{unsafe}}} \left[ 1 - \texttt{sigmoid}\left(-u(\sigma)\right) \right] \tag{8}$$

The offline dataset $D$ is divided into two subdataset. $D_{\text{safe}}$ contains all the safe segments while all the unsafe segments are in $D_{\text{unsafe}}$. The weights $\lambda_s$ and $\lambda_u$ are determined by the ratio of the number of safe segments $n_s$ to the number of unsafe segments $n_u$ in the offline dataset, specifically, $\frac{\lambda_s n_s}{\lambda_u n_u} = \eta$. The hyperparameter $\eta$ regulates the relative importance of safe and unsafe segments. By optimizing Equation 8, we assign high scores to safe segments and low scores to unsafe ones. Accordingly, based on the formulation of $\psi_{\pi_\theta}(\sigma)$ in Equation 7, a policy can be learned that generates safe trajectories with high probability while avoiding unsafe behavior. We also show the following result to justify our choice of trajectory score and the loss function. The proof is in appendix A.

---

[2]To maintain consistency, we use the same $\theta$ as in the preference alignment module.

**Lemma 5.1.** *Minimizing the loss function in Equation 8 increases the log probability of safe trajectories in the safe dataset $D_{safe}$ and decreases the log probability of trajectories in the unsafe dataset $D_{unsafe}$ under policy $\pi$.*

## 5.2 INTEGRATING PREFERENCE AND SAFETY ALIGNMENT (PRESA)

From the above discussion, we can learn a policy that aligns with human preferences using pairwise comparisons or adheres to implicit safety constraints using binary safety labels. However, integrating both safety and reward preferences is required in our setting. Combining these two objectives in an ad-hoc manner may not result in a stable and robust method. Therefore, we propose a method to integrate these two learning modules in a principled manner into a single objective function that satisfies both criteria. We refer to our approach as *PreSa*.

### 5.2.1 SAFETY ALIGNMENT AS DEFINING FEASIBLE POLICY SET

The two components in Equation 8 address safe and unsafe segments separately, corresponding to the safety labels for each segment. We can rewrite the equation by combining these two components using the safety labels:

$$L_{\text{safety}}(\pi_\theta, D) = \mathbb{E}_{\sigma \sim D}\left[w(y_\sigma)(1 - \texttt{sigmoid}\left(y_\sigma(\psi_{\pi_\theta}(\sigma) - z_{\text{ref}})\right))\right] \tag{9}$$

where $y_\sigma$ is the safety label of the corresponding segment $\sigma$ and $w(y_\sigma)$ is the weights for safe and unsafe segments respectively:

$$w(y_\sigma) = \begin{cases} \lambda_s & \text{if } y_\sigma = +1 \\ \lambda_u & \text{if } y_\sigma = -1 \end{cases} \tag{10}$$

Let us temporally ignore the weights $w(y_\sigma)$ as they are irrelevant to this discussion, and focus on the latter term in Equation 9. We observe that $y_\sigma(\psi_{\pi_\theta}(\sigma) - z_{\text{ref}})$ serves as a scalar score of segment $\sigma$ and the `sigmoid` function can be interpreted as providing the probability of predicting the label $y_\sigma$ for the corresponding segment $\sigma$ under the policy $\pi_\theta$,

$$p(Y = y_\sigma|\sigma; \pi_\theta) \triangleq \texttt{sigmoid}\left(y_\sigma(\psi_{\pi_\theta}(\sigma) - z_{\text{ref}})\right) \tag{11}$$

Therefore, in the context of a typical binary classification problem, minimizing the loss function in Equation 9 is equivalent to maximizing the probability of correctly classifying each segment with respect to safety. This classification-like objective determines which segments are safe, and which are unsafe. Consequently, we find that the safety alignment objective can be transformed to define a feasible policy set as follows,

$$\Pi = \{\pi|p(Y = y_\sigma|\sigma; \pi) \geq \delta, \forall \sigma\} \tag{12}$$

where $\delta$ is a predefined parameter that controls the stringency with which we accept a segment as being correctly classified according to the safety labels provided by humans. When $\delta$ is close to 1, we expect each segment to be classified correctly with a very high probability. In this case, if the ground truth label is $+1$, the segment's score should be high, and the learned policy should assign high probabilities to this segment according to Equation 7. If the ground truth label is $-1$, the segment's score should be low, and the learned policy should assign low probabilities to it. When $\delta$ is more relaxed, we allow for greater tolerance, which may lead to a policy that occasionally violates the implicit safety constraints.

### 5.2.2 UNIFIED OBJECTIVE OF PRESA

As the preference alignment module ensures consistency with human preferences and the safety alignment module implicitly defines a feasibility policy set, we can now integrate them into a single objective function within a constrained optimization framework. The unified objective is presented as follows,

$$\min_{\pi_\theta} \mathbb{E}_{(\sigma^+, \sigma^-) \sim D}\left[-\log P_{\pi_\theta}\left[\sigma^+ \succ \sigma^-\right]\right]$$

$$\text{s.t., } \mathbb{E}_{(\sigma, y_\sigma) \sim D}\left[p(Y = y_\sigma|\sigma; \pi_\theta)\right] \geq \delta \tag{13}$$

where the objective term for reward preference is defined analogous to the CPL objective in Equation 2 without explicitly learning the rewards. Compared to the objective function of safe reinforcement

learning in Equation 1, the term of minimizing negative log-likelihood of preferences in Equation 13 corresponds to the maximization of cumulative reward. The feasible policy set determined by the safety alignment module corresponds to the one defined by the cumulative cost constraints. Given the above constrained objective, we expect to learn a policy that maximizes reward while adhering to the safety constraints implicitly encoded in the safety labels. Notably, PreSa does not require learning of reward and cost functions, and avoids doing constrained RL. To solve this constrained optimization problem, we employ the Lagrangian method to convert the constrained primal problem into an unconstrained dual form:

$$\min_{\pi_\theta} \max_{\nu \geq 0} L(\pi_\theta, \nu, D) = \mathbb{E}_{(\sigma^+,\sigma^-)\sim D} \left[ -\log P_{\pi_\theta} \left[ \sigma^+ \succ \sigma^- \right] \right] + \nu \cdot \left( \delta - \mathbb{E}_{(\sigma,y_\sigma)\sim D} \left[ p(Y = y_\sigma | \sigma; \pi_\theta) \right] \right)$$
(14)

where $\nu \geq 0$ is the Lagrange multiplier. By introducing the weights $w(y_\sigma)$ back, the above objective function is rewritten as,

$$L(\pi_\theta, \nu, D) = \mathbb{E}_{(\sigma^+,\sigma^-)\sim D} \left[ -\log P_{\pi_\theta} \left[ \sigma^+ \succ \sigma^- \right] \right] + \nu \cdot \mathbb{E}_{(\sigma,y_\sigma)\sim D} \left[ w(y_\sigma) \cdot (\delta - p(Y = y_\sigma | \sigma; \pi_\theta)) \right]$$
(15)

Interestingly, the optimization of preference learning may sometimes conflict with the objective of optimizing safety alignment, though they can also complement each other in learning a better policy. That is because preference and safety do not influence each other in a monotonic manner. For instance, some unsafe segments might be preferred while some unpreferred segments could be safe as well. Thus, in Equation 15, the Lagrange multiplier $\nu$ dynamically manages the mutual influence. To address the optimization problem, we iteratively update the policy parameter $\theta$ and the Lagrange multiplier $\nu$ using gradient descent, which helps avoid over-emphasizing one objective at the expense of the other due to a fixed optimization ratio.

## 6 EXPERIMENTS

In this section, we present experiments to evaluate the effectiveness of PreSa in achieving both high reward performance and adherence to safety constraints. We aim to address the following questions:

- How does PreSa compare to Offline Safe POHF baselines and those from offline safe RL using ground truth reward and cost?
- How do the preference alignment and safety alignment modules perform individually?
- How does PreSa perform with varying trajectory segment lengths and different offline dataset sizes?
- What ingredients of PreSa are important for enhanced performance and safety alignment?

### 6.1 EXPERIMENTAL SETTING

We conduct our experiments using the well-established DSRL benchmark (Liu et al., 2023a) which is designed for offline safe RL, leveraging it to generate synthetic human feedback for evaluation. This benchmark provides pre-collected offline data across 29 tasks in two widely used domains: SafetyGym (Ray et al., 2019; Ji et al., 2023) and BulletGym (Gronauer, 2022). These tasks feature various agents aiming to achieve high rewards or goals while avoiding obstacles or maintaining a predefined safe velocity. To ensure fair comparison and evaluation, we adopt the constraint variation evaluation method introduced in DSRL (Liu et al., 2023a). Each method is tested on every task using three different cost thresholds and three random seeds to ensure consistency. Evaluation metrics include normalized reward and normalized cost, where a cost below 1 signifies safety (Liu et al., 2023a; Fu et al., 2020).

For the practical implementation of PreSa, we follow a similar training pipeline to CPL (Hejna et al., 2024). Initially, the policy is pretrained using behavior cloning (BC) with the offline dataset, which is then retained as the reference policy, $\pi_{\text{ref}}$. Following this, the policy is optimized with PreSa using human pairwise preferences and binary safety labels.

**Synthetic Human Feedback** For evaluation, we generate synthetic human feedback from the offline dataset provided by DSRL. We first randomly select 10,000 pairs of trajectory segments with varying lengths from each dataset, which allows for testing with fewer feedbacks via random sampling.

Pairwise trajectory segments are then labeled using the ground truth rewards from the original offline dataset. We base pairwise preferences on cumulative rewards rather than estimated advantages from a trained policy, as in the dense reward setting, the optimal advantage is essentially a reshaped version of the reward, leading to the same optimal policy (Hejna et al., 2024). Additionally, we label trajectory segments for safety using ground truth cost values and predefined safety thresholds. In line with the experimental settings in DSRL, these safety thresholds apply to the entire trajectories. When labeling trajectory segments, we use a reshaped threshold that is adjusted proportionally based on the length of the segment relative to the maximum trajectory length within each domain.

**Baselines** We compare our approach against several baselines to demonstrate its effectiveness, which are across Offline Safe POHF setting and offline safe RL setting with ground truth rewards and costs. In Offline Safe POHF setting, we consider three baseline methods: 1) *Binary Alignment*: Inspired by Ethayarajh et al. (2024), unified binary labels are generated based on comparative preferences and binary safety labels. Segments that are both safe and preferred are assigned a label of $+1$, while all others $-1$. The policy is then learned solely using the safety alignment module. 2) *BC-Safe-Seg*: A behavior cloning (BC) approach trained only on safe trajectory segments. 3) *Safe-RLHF (CDT)* (Dai et al., 2024): A variant of Safe-RLHF adapted to our setting where reward and cost models are learned from human feedback and a state-of-the-art offline safe RL approach, CDT Liu et al. (2023b) is applied for policy optimization. Additionally, we select three baselines from offline safe RL setting. They assume access to ground truth reward and cost: 1) *BC-All*: BC trained on the entire dataset. 2) *BC-Safe*: BC trained exclusively on safe trajectories that meet safety constraints. 3) *CDT*: A sequence modeling approach that incorporates safety constraints into Decision Transformers.

## 6.2 RESULTS

**How Does PreSa Perform?** Extensive experiments were conducted to evaluate PreSa's performance against baselines across all 29 tasks within SafetyGym and BulletGym. Figure 2 shows the proportion of safe policies learned by each approach in both domains. PreSa surpasses all Offline Safe POHF baselines with more safe policies learned. Compared to offline safe RL baselines, PreSa performs comparably in SafetyGym, outperforming two baselines by a significant margin, except for BC-Safe, while in BulletGym, PreSa outperforms all baselines with most safe policies learned. Notably, the offline safe RL baselines have access to ground truth data, while PreSa relies solely on human feedback which has much less information. Despite this, PreSa matches or exceeds the performance of these baselines, highlighting its effectiveness.

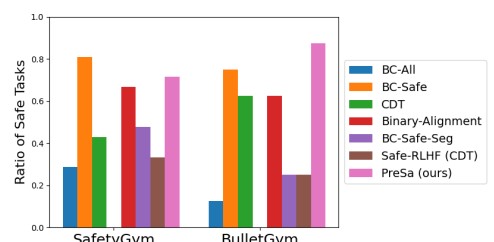

Figure 2: Ratio of safe agents learned by different approaches.

The complete results using normalized rewards and costs are presented in Table 1. Among the Offline Safe POHF baselines, Binary Alignment achieves low costs but also results in very low rewards across all tasks. This occurs because $+1$ is assigned to safe but often low-reward segments, as they are only marginally better than their counterparts. BC-Safe-Seg achieves relatively high average rewards, but this is largely due to unsafe agents. Safe-RLHF (CDT) manages to learn higher rewards but struggles with safe policy learning. PreSa outperforms all Offline Safe POHF baselines by successfully adhering to implicit safety constraints while also achieving high rewards. Similarly, although the offline safe RL baselines achieve high rewards, they struggle to learn safe behaviors. In contrast, PreSa shows significantly better performance in learning safe policies.

**Individual Performance of Preference and Safety Alignment Modules** We illustrate the results for all tasks within the BulletGym domain in Figure 3. The preference alignment module is designed to learn a policy that aligns with human pairwise preferences, implicitly capturing reward information without accounting for safety. Consequently, the policy learned by this module tends to achieve relatively high rewards. In contrast, the safety alignment module leverages binary safety labels, which encode implicit safety constraints, and successfully learns safe policies for most tasks. The effectiveness of each alignment module forms a solid foundation for learning high-reward, safe

Table 1: All evaluation results of normalized reward and cost. The ↑ symbol indicates that the higher reward, the better, while the ↓ symbol signifies that the lower cost (up to a threshold of 1), the better. **Bold**: Safe agents whose normalized cost is below 1. **Blue**: Safe agent with the highest reward among offline safe RL baselines. **Orange**: Safe agent with the highest reward among approaches learning from offline synthetic human feedback.

| Task | BC-All | | BC-Safe | | CDT | | Binary Alignment | | BC-Safe-Seg | | Safe-RLHF (CDT) | | PreSa (Ours) | |
|---|---|---|---|---|---|---|---|---|---|---|---|---|---|---|
| | reward↑ | cost↓ | reward↑ | cost↓ | reward↑ | cost↓ | reward↑ | cost↓ | reward↑ | cost↓ | reward↑ | cost↓ | reward↑ | cost↓ |
| PointButton1 | 0.1 | 1.05 | 0.06 | 0.52 | 0.53 | 1.68 | 0.02 | 0.55 | 0.06 | 0.81 | 0.06 | 0.78 | 0.09 | 0.84 |
| PointButton2 | 0.27 | 2.02 | 0.16 | 1.1 | 0.46 | 1.57 | -0.03 | 0.5 | 0.22 | 1.57 | 0.18 | 1.33 | -0.1 | 0.74 |
| PointCircle1 | 0.79 | 3.98 | 0.41 | 0.16 | 0.59 | 0.69 | -0.23 | 1.21 | 0.32 | 1.09 | 0.37 | 2.97 | 0.4 | 0.21 |
| PointCircle2 | 0.66 | 4.17 | 0.48 | 0.99 | 0.64 | 1.05 | -0.24 | 8.3 | 0.44 | 1.89 | 0.66 | 4.87 | 0.16 | 0.96 |
| PointGoal1 | 0.65 | 0.95 | 0.43 | 0.54 | 0.69 | 1.12 | 0.31 | 0.42 | 0.48 | 1.17 | 0.34 | 0.52 | 0.37 | 0.73 |
| PointGoal2 | 0.54 | 1.97 | 0.29 | 0.78 | 0.59 | 1.34 | 0.39 | 1.15 | 0.52 | 2.08 | 0.35 | 2.5 | 0.16 | 0.96 |
| PointPush1 | 0.19 | 0.61 | 0.13 | 0.43 | 0.24 | 0.48 | 0.14 | 0.51 | 0.19 | 0.6 | 0.1 | 0.36 | 0.14 | 0.4 |
| PointPush2 | 0.18 | 0.91 | 0.11 | 0.8 | 0.21 | 0.65 | 0.17 | 0.8 | 0.18 | 0.8 | 0.08 | 0.22 | 0.12 | 0.9 |
| CarButton1 | 0.03 | 1.38 | 0.07 | 0.85 | 0.21 | 1.6 | -0.01 | 2.52 | 0.02 | 1.42 | 0.05 | 3.96 | 0.12 | 1.87 |
| CarButton2 | -0.13 | 1.24 | -0.01 | 0.63 | 0.13 | 1.58 | -0.06 | 1.36 | -0.03 | 1.01 | 0.02 | 1.77 | -0.04 | 1.27 |
| CarCircle1 | 0.72 | 4.39 | 0.37 | 1.38 | 0.6 | 1.73 | -0.32 | 4.71 | 0.61 | 4.53 | 0.27 | 3.53 | -0.26 | 2.86 |
| CarCircle2 | 0.76 | 6.44 | 0.54 | 3.38 | 0.66 | 2.53 | -0.23 | 0.0 | 0.63 | 4.23 | 0.5 | 3.91 | 0.23 | 0.22 |
| CarGoal1 | 0.39 | 0.33 | 0.24 | 0.28 | 0.66 | 1.21 | 0.29 | 0.38 | 0.25 | 0.3 | 0.4 | 0.61 | 0.26 | 0.14 |
| CarGoal2 | 0.23 | 1.05 | 0.14 | 0.51 | 0.48 | 1.25 | 0.18 | 0.64 | 0.17 | 1.03 | 0.18 | 1.01 | 0.14 | 0.35 |
| CarPush1 | 0.22 | 0.36 | 0.14 | 0.33 | 0.31 | 0.4 | 0.16 | 0.34 | 0.21 | 0.51 | 0.17 | 0.96 | 0.15 | 0.56 |
| CarPush2 | 0.14 | 0.9 | 0.05 | 0.45 | 0.19 | 1.3 | 0.07 | 0.69 | 0.07 | 0.91 | 0.1 | 1.81 | 0.1 | 0.52 |
| SwimmerVelocity | 0.49 | 4.72 | 0.51 | 1.07 | 0.66 | 0.96 | -0.04 | 0.7 | 0.33 | 2.61 | 0.66 | 1.1 | 0.39 | 1.96 |
| HopperVelocity | 0.65 | 6.39 | 0.36 | 0.67 | 0.63 | 0.61 | -0.02 | 0.0 | 0.64 | 0.64 | 0.17 | 1.27 | 0.42 | 5.89 |
| HalfCheetahVelocity | 0.97 | 13.1 | 0.88 | 0.54 | 1.0 | 0.01 | 0.05 | 0.0 | 0.92 | 0.54 | 0.93 | 0.37 | 0.71 | 4.11 |
| Walker2dVelocity | 0.79 | 3.88 | 0.79 | 0.04 | 0.78 | 0.06 | -0.01 | 0.0 | 0.78 | 0.01 | 0.11 | 1.42 | 0.79 | 0.0 |
| AntVelocity | 0.98 | 3.72 | 0.98 | 0.29 | 0.98 | 0.39 | -0.06 | 0.0 | 0.96 | 0.3 | 0.93 | 0.23 | 0.96 | 0.27 |
| **SafetyGym Average** | 0.46 | 3.03 | 0.34 | 0.75 | 0.54 | 1.06 | 0.03 | 1.22 | 0.38 | 1.34 | 0.32 | 1.7 | 0.25 | 1.23 |
| BallRun | 0.6 | 5.08 | 0.27 | 1.46 | 0.39 | 1.16 | 0.31 | 4.79 | 0.37 | 1.13 | 0.35 | 1.65 | 0.19 | 0.09 |
| CarRun | 0.97 | 0.33 | 0.94 | 0.22 | 0.99 | 0.65 | 0.94 | 0.0 | 0.97 | 0.95 | 0.87 | 1.16 | 0.96 | 0.0 |
| DroneRun | 0.24 | 2.13 | 0.28 | 0.74 | 0.63 | 0.79 | 0.11 | 0.17 | 0.17 | 5.97 | 0.47 | 3.12 | 0.16 | 0.33 |
| AntRun | 0.72 | 2.93 | 0.65 | 1.09 | 0.72 | 0.91 | 0.09 | 0.01 | 0.66 | 1.38 | 0.72 | 1.04 | 0.61 | 0.63 |
| BallCircle | 0.74 | 4.71 | 0.52 | 0.65 | 0.77 | 1.07 | 0.06 | 0.24 | 0.39 | 0.68 | 0.68 | 1.2 | 0.22 | 0.03 |
| CarCircle | 0.58 | 3.74 | 0.5 | 0.84 | 0.75 | 0.95 | 0.06 | 0.35 | 0.56 | 1.76 | 0.57 | 0.84 | 0.08 | 0.91 |
| DroneCircle | 0.72 | 3.03 | 0.56 | 0.57 | 0.63 | 0.98 | -0.23 | 1.59 | 0.61 | 1.9 | 0.61 | 0.87 | 0.54 | 0.72 |
| AntCircle | 0.58 | 4.9 | 0.4 | 0.96 | 0.54 | 1.78 | 0.48 | 3.03 | 0.54 | 3.15 | 0.45 | 2.04 | 0.55 | 3.78 |
| **BulletGym Average** | 0.64 | 3.36 | 0.52 | 0.82 | 0.68 | 1.04 | 0.23 | 1.27 | 0.53 | 2.11 | 0.59 | 1.49 | 0.41 | 0.81 |

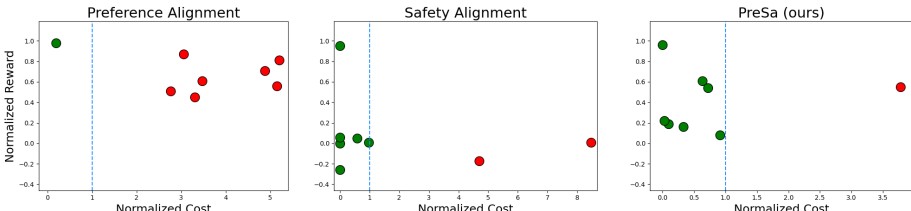

Figure 3: Visualization of normalized reward and cost for each task within BulletGym domain. The dotted blue vertical lines mark the cost threshold of 1. Each round dot represents a task, where green dots indicate tasks meeting safety constraints, and red dots indicate tasks with constraint violations.

policies. When both modules are integrated, PreSa refines the preference-aligned policy to simultaneously prioritize safer behaviors.

**PreSa Performance With Different Trajectory Segment Lengths, Dataset Sizes** To further investigate how PreSa performs in different settings, we evaluate its effectiveness compared to baselines across varying segment lengths and offline dataset sizes. The results of DroneCircle task, shown in Figure 4, indicate that as the length of the segments increases, PreSa consistently learns policies with lower costs while maintaining stable reward performance, compared to other baselines. Moreover, PreSa demonstrates a stable and better performance as more offline data becomes available. These results show that PreSa outperforms other baselines across different segment lengths and dataset sizes.

**What Contributes To PreSa's Performance?** Ablation studies are conducted to assess the impact of varying the hyperparameters $\alpha$ and $\beta$, which are used by the preference alignment and safety alignment modules, respectively, to regularize the learning policy with a pretrained reference policy. The results for BallRun and DroneCircle tasks are shown in Table 3. We find that PreSa remains robust across different selections of $\alpha$ and $\beta$, although higher performance could likely be achieved with further hyperparameter tuning.

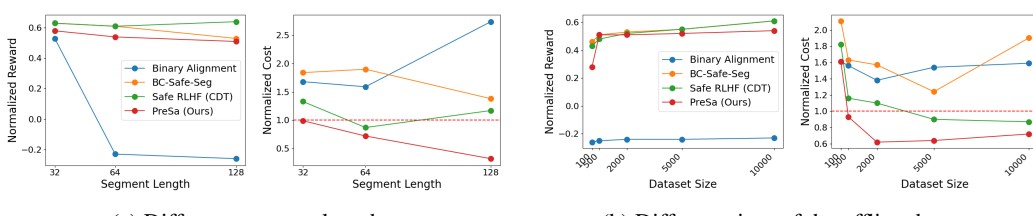

(a) Different segment lengths.      (b) Different sizes of the offline dataset.

Figure 4: Performance of Offline Safe POHF methods across varying trajectory segment lengths, dataset sizes.

Table 3: Ablation study of varying values of $\alpha$ and $\beta$.

| Task | $\alpha = 0.2$ | | $\alpha = 0.4$ | | $\alpha = 0.6$ | | $\alpha = 0.8$ | | $\beta = 0.25$ | | $\beta = 0.5$ | | $\beta = 0.75$ | | $\beta = 1.0$ | |
|---|---|---|---|---|---|---|---|---|---|---|---|---|---|---|---|---|
| | reward↑ | cost↓ | reward↑ | cost↓ | reward↑ | cost↓ | reward↑ | cost↓ | reward↑ | cost↓ | reward↑ | cost↓ | reward↑ | cost↓ | reward↑ | cost↓ |
| BallRun | 0.19 | 0.09 | 0.19 | 0.1 | 0.38 | 2.8 | 0.19 | 0.12 | 0.18 | 0.11 | 0.18 | 0.09 | 0.34 | 2.67 | 0.19 | 0.09 |
| DroneCircle | 0.54 | 0.72 | 0.5 | 0.92 | 0.27 | 1.9 | 0.54 | 1.09 | 0.52 | 1.04 | 0.49 | 0.98 | 0.32 | 1.45 | 0.54 | 0.72 |

We also investigate the effect of the reference point $z_{\text{ref}}$, which is used to determine the "gain" or "loss" of a trajectory segment based on the relative difference between the segment's score and the reference point. In our approach, $z_{\text{ref}}$ is an estimated average score for all segments. To demonstrate its effectiveness, we performed an ablation study on $z_{\text{ref}}$, as shown in Table 2. The results indicate that without the reference point, PreSa's performance drops significantly, making it difficult to find safe policies. This occurs because, without the reference point, the utility of a segment is based solely on its absolute score rather than a relative value, which destabilizes the learning process and degrades performance.

Table 2: Ablation study of $z_{\text{ref}}$.

| Task | w/o $z_{\text{ref}}$ | | PreSa | |
|---|---|---|---|---|
| | reward↑ | cost↓ | reward↑ | cost↓ |
| BallRun | 0.29 | 1.59 | **0.19** | **0.09** |
| CarRun | **0.95** | **0.0** | 0.96 | 0.0 |
| DroneRun | 0.61 | 2.77 | **0.16** | **0.33** |
| AntRun | 0.67 | 2.68 | **0.61** | **0.63** |
| BallCircle | **0.39** | **0.67** | 0.22 | 0.03 |
| CarCircle | **0.46** | **0.96** | 0.08 | 0.91 |
| DroneCircle | 0.32 | 1.52 | **0.54** | **0.72** |
| AntCircle | 0.59 | 5.67 | 0.55 | 3.78 |
| Average | 0.53 | 1.93 | **0.41** | **0.81** |

## 7 CONCLUSION

In this paper, we present Offline Safe POHF, a framework in which a policy is learned using human pairwise preferences and binary safety labels for each trajectory segment, without access to ground truth rewards and costs. We first analyze the problem through two distinct modules: preference alignment and safety alignment, which can be applied individually for policy learning based on preferences or safety labels, respectively. We then introduce an approach for Offline Safe POHF problems, called PreSa, which integrates both modules into a single constrained optimization objective, as the safety alignment module implicitly defines a feasible set of policies. Given offline human feedback data, PreSa directly learns a policy without the need to develop additional reward and cost models or employ constrained safe RL. Empirical results demonstrate that PreSa not only outperforms baselines from the Offline Safe POHF setting but also matches or surpasses offline safe RL approaches that assume access to ground truth rewards and costs.

**Limitations** For evaluation, we generate synthetic human feedback based on the offline safe RL benchmark, DSRL. While utilizing synthetic human feedback is a common practice in PbRL research (Christiano et al., 2017; Lee et al., 2021; Park et al., 2022), and synthetic labels can effectively approximate real human feedback under certain conditions (Metcalf et al., 2024), it is important to note that scripted human labelers may not achieve perfect consistency with actual human labelers. Future work will involve conducting extended experiments with real human participants.

### REPRODUCIBILITY STATEMENT

We include the necessary proofs and detailed experimental settings in the Appendix to facilitate the reproduction of the experiments and results presented in this paper.

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

## A SCORE FUNCTION JUSTIFICATION

We first make an assumption that discount factor $\gamma$ is close to 1, which is often the case in practice.

**Lemma A.1.** *Minimizing the loss function in Equation 8 increases the log probability of safe trajectories in the safe dataset $D_{safe}$ and decreases the log probability of trajectories in the unsafe dataset $D_{unsafe}$ under policy $\pi$.*

*Proof.* For simplicity, we ignore the $\beta$ term from Equation 7. We have:

$$u(\sigma) \approx \sum_{t=0}^{T} \log \pi(a_t|s_t) - \sum_{t=0}^{T} \log \pi_{\text{ref}}(a_t|s_t) - z_{\text{ref}}(\pi) \tag{16}$$

The approximation sign is due to ignoring the discount factor (being close to 1); $z_{\text{ref}}(\pi)$ is defined as the average over all trajectories (both safe, unsafe): $\mathbb{E}_\sigma \left[ \sum_{t=0}^{T} \gamma^t \beta \log \frac{\pi(a_t|s_t)}{\pi_{\text{ref}}(a_t|s_t)} \right]$.

The log progability of a trajectory $\sigma$ as per policy $\pi$ is given as:

$$\log p(\sigma; \pi) = \sum_{t=0}^{T} \log \pi(a_t|s_t) + \text{constants} \tag{17}$$

where constant terms refer to the log of transition function, which is independent of $\pi$. Using Equation 16, we have:

$$\log p(\sigma; \pi) \approx u(\sigma) + \sum_{t=0}^{T} \log \pi_{\text{ref}}(a_t|s_t) + z_{\text{ref}}(\pi) + \text{constants}$$

$$= u(\sigma) + z_{\text{ref}}(\pi) + \text{constants} \tag{18}$$

Notice that $z_{\text{ref}}(\pi)$ only depends on policy $\pi$ and is the same for all the trajectories in the safe and unsafe datasets. Minimizing the loss function in Equation 8 would optimize the policy $\pi_\theta$ such that higher score $u$ is assigned to safe trajectories $\sigma^+$ and lower scores to $\sigma^-$. Thus, as per Equation 18, log probabilities of safe trajectories would tend to increase and unsafe trajectories would tend to decrease. $\square$

## B EXPERIMENTAL DETAILS

This section provides the experimental details required to reproduce the experiments and results presented in our paper.

### B.1 TASK DESCRIPTION

We conducted our experiments and generated synthetic human feedback using the well-established DSRL benchmark Liu et al. (2023a), which offers datasets specifically designed for offline safe RL research. This benchmark includes 29 datasets with various safe RL environments and difficulty levels in SafetyGym Ray et al. (2019); Ji et al. (2023) and BulletGym Gronauer (2022).

- **SafetyGym** is a suite of environments built on the Mujoco physics simulator, with a diverse set of tasks. It features two types of agents, `Car` and `Point`, each tasked with four different activities: `Button`, `Circle`, `Goal`, and `Push`. The difficulty of these tasks is further distinguished by levels, denoted by `1` and `2`. In each task, the agents must reach a goal while avoiding hazards and obstacles. Moreover, SafetyGym includes five velocity-constrained tasks for different agents: `Ant`, `HalfCheetah`, `Hopper`, `Walker2d`, and `Swimmer`. Figure 5a illustrates the agents and tasks within SafetyGym.

- **BulletGym** is a collection of environments built using the PyBullet physics simulator. Similar to SafetyGym, it focuses on safety-critical tasks but has shorter time horizons and a wider variety of agents. The suite includes four types of agents: `Ball`, `Car`, `Drone`, and `Ant`, each with two tasks: `Circle` and `Run`. The agents and tasks within BulletGym are shown in Figure 5b.

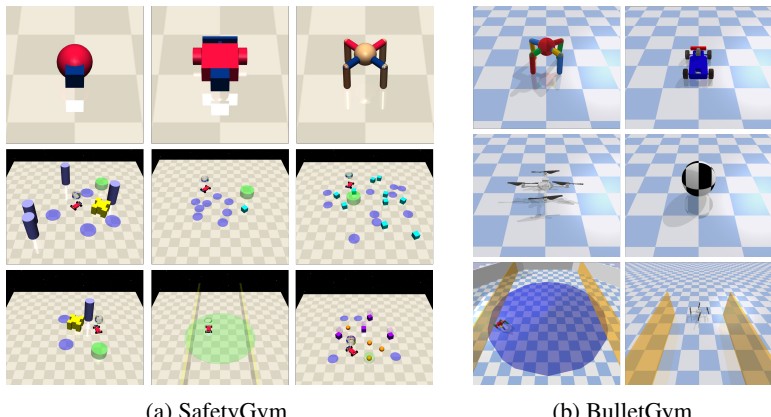

(a) SafetyGym       (b) BulletGym

Figure 5: Visualization of agents and tasks in SafetyGym and BulletGym.

## B.2    SYNTHETIC HUMAN FEEDBACK

To evaluate our approach in the Offline Safe POHF setting, we generate synthetic human feedback consisting of pairwise preferences regarding agent behavior and safety labels indicating whether the agent's actions are safe or not. For pairwise preferences, we follow the method used in CPL (Hejna et al., 2024), assuming that the human model follows a regret-based preference framework. Given that negated regret is equivalent to the discounted sum of optimal advantages, we provide preference feedback based on the cumulative advantages of the compared trajectory segments. However, since obtaining optimal advantages requires running RL for each task, we simplify the process by using cumulative rewards instead. This is because optimal advantages are essentially a reshaped version of the reward, leading to the same optimal policy.

For the binary safety labels, we use the ground truth cost data from the original offline dataset to assess the safety of each trajectory segment. Safety labels are then assigned based on predefined cost thresholds, which are generally defined for the entire task or trajectory. To label individual segments (which are parts of full trajectories), we proportionally adjust the cost threshold according to the segment's length relative to the maximum trajectory length in each domain.

## B.3    EVALUATION METRICS

To assess the algorithm's performance, we adopt the evaluation methodology from DSRL (Liu et al., 2023a), using normalized reward and normalized cost as metrics. The normalized reward is defined as follows:

$$R_{\text{normalized}} = \frac{R_\pi - r_{\min}}{r_{\max} - r_{\min}}$$

where $R_\pi$ is the cumulative reward under policy $\pi$, and $r_{\max}$ and $r_{\min}$ denote the empirical maximum and minimum reward returns. The normalized cost is represented as:

$$C_{\text{normalized}} = \frac{C_\pi + \epsilon}{\kappa + \epsilon}$$

where $C_\pi$ is the cumulative cost under policy $\pi$. The cost threshold is given by $\kappa$, and $\epsilon$ is a small positive constant added to ensure numerical stability when $\kappa = 0$. According to the DSRL benchmark, a task is considered safe if the normalized cost does not exceed 1.

## B.4    TRAINING DETAILS AND HYPERPARAMETERS

Our approach follows a two-step training process. In the first step, we pretrain the policy using behavior cloning (BC) on all trajectory segments, establishing a reference policy, denoted as $\pi_{\text{ref}}$, which will later regulate the learning policy $\pi$. In the second step, we optimize the policy by applying PreSa with offline human feedback.

Table 4: Hyperparameters of PreSa for tasks in two domains.

| Hyperparameters | BulletGym | SafetGym |
|---|---|---|
| Training Steps | 100k | 200k |
| Pretraining Steps | 30k | 60k |
| Batch Size | 32 | 96 |
| Policy network architecture | [256, 256] MLP | [256, 256] MLP |
| Policy network dropout | 0.1 | 0.25 |
| Optimizer | Adam | Adam |
| Policy learning Rate | 0.0001 | 0.0001 |
| Temperature $\alpha$ | 0.2 | 0.2 |
| Temperature $\beta$ | 1.0 | 0.2 |
| Discount factor $\gamma$ | 1.0 | 1.0 |
| Balancing factor $\eta$ | 0.1 | 2.0 |
| Constraint lower bound $\delta$ | 0.95 | 0.9 |
| Lagrange multiplier learning rate | 0.005 | 0.0005 |

The hyperparameters used in the experiments are summarized in Table 4. We assume all policies to be Gaussian with a fixed variance. Actions are predicted using a standard multi-layer perceptron (MLP), and the log probability $\log \pi(a|s)$ is calculated as $-|\pi(s) - a|_2^2$, following the implementation design from CPL. The policy networks are structured with two hidden layers, each containing 256 hidden units and employing ReLU activation functions, with dropout applied. Different values for the hyperparameters in PreSa are used depending on the specific domain.

## C    EXTENDED RESULTS

In this section, we provide supplementary results, which include PreSa's performance across different segment lengths and dataset sizes, comprehensive ablation studies on several main hyperparameters (such as $\alpha$, $\beta$, and $\eta$), and the learning curves of PreSa for all 29 tasks.

### C.1    PERFORMANCE OF PRESA WITH VARYING SEGMENT LENGTHS AND DATASET SIZES

The comprehensive results of PreSa with varying segment lengths are presented in Table 5. Overall, PreSa demonstrates strong performance across different segment lengths. However, for certain specific tasks, such as DroneRun and BallCircle, the policies learned under different segment lengths exhibit varying performance.

Table 6 presents the results of PreSa trained with different sizes of offline datasets. Unsurprisingly, when trained on only 100 pairs of trajectory segments, PreSa struggles to learn safe policies. However, as the size of the offline datasets increases, we observe that PreSa trained on a few hundred to thousands segment pairs can achieve performance comparable to that of models trained on 10,000 pairs. This suggests the effectiveness of PreSa.

### C.2    PERFORMANCE OF PRESA WITH IMPERFECT SAFETY FEEDBACK

To evaluate the effectiveness of PreSa under imperfect safety feedback, we simulate different levels of imperfection by introducing noise into the safety feedback. At each level, a subset of the feedback is randomly selected, and its True/False labels are flipped. The results, shown in Table 7, demonstrate that PreSa consistently outperforms the baselines across varying levels of imperfect safety feedback, although its performance gradually declines as the noise level increases.

### C.3    ADDITIONAL BASELINES

We conducted experiments on additional baselines, with the results presented in Table 8 and Table 9. *Unified Comparison*: Unified pairwise preferences were used, where high-reward and safe trajectories were preferred. *Safe-RLHF (CDT) (Cost: binary label only)*: A variant of Safe-RLHF (CDT) in

Table 5: Performance of PreSa with varying segment lengths

| Segment Length | 32 | | 64 | | 128 | |
|---|---|---|---|---|---|---|
| | reward↑ | cost↓ | reward↑ | cost↓ | reward↑ | cost↓ |
| BallRun | **0.26** | **1.12** | **0.19** | **0.09** | - | - |
| CarRun | **0.96** | **0.0** | **0.96** | **0.0** | 0.96 | 0.0 |
| DroneRun | **0.17** | **0.18** | 0.16 | 0.33 | -0.01 | 0.0 |
| AntRun | **0.64** | **0.79** | 0.61 | 0.63 | 0.69 | 1.26 |
| BallCircle | **0.36** | **0.25** | 0.22 | 0.03 | 0.22 | 0.05 |
| CarCircle | 0.11 | 1.29 | **0.08** | **0.91** | 0.07 | 0.79 |
| DroneCircle | **0.58** | **0.99** | 0.54 | 0.72 | 0.51 | 0.32 |
| AntCircle | 0.56 | 3.88 | 0.55 | 3.78 | 0.57 | 3.88 |
| **BulletGym Average** | 0.45 | 1.06 | **0.41** | **0.81** | **0.43** | **0.9** |

Table 6: Performance of PreSa with varying offline dataset sizes

| Offline Data | 100 pairs | | 500 pairs | | 2000 pairs | | 5000 pairs | | 10000 pairs | |
|---|---|---|---|---|---|---|---|---|---|---|
| | reward↑ | cost↓ | reward↑ | cost↓ | reward↑ | cost↓ | reward↑ | cost↓ | reward↑ | cost↓ |
| BallRun | 0.27 | 1.67 | **0.21** | **0.23** | **0.19** | **0.14** | **0.19** | **0.14** | **0.19** | **0.09** |
| CarRun | **0.91** | **0.0** | **0.96** | **0.0** | **0.96** | **0.0** | **0.96** | **0.0** | **0.96** | **0.0** |
| DroneRun | 0.38 | 5.11 | 0.2 | 2.6 | **0.19** | **0.52** | 0.17 | 1.24 | **0.16** | **0.33** |
| AntRun | 0.5 | 1.72 | **0.61** | **0.95** | **0.64** | **0.88** | 0.59 | 0.61 | **0.61** | **0.63** |
| BallCircle | **0.26** | **0.77** | 0.24 | 0.07 | 0.21 | 0.06 | 0.2 | 0.02 | 0.22 | 0.03 |
| CarCircle | 0.26 | 1.55 | **0.17** | **0.85** | **0.1** | **0.56** | 0.09 | 1.18 | 0.08 | 0.91 |
| DroneCircle | 0.28 | 1.61 | **0.51** | **0.93** | **0.51** | **0.62** | **0.52** | **0.64** | 0.54 | 0.72 |
| AntCircle | 0.5 | 3.67 | 0.52 | 3.09 | 0.56 | 3.91 | 0.57 | 3.86 | 0.55 | 3.78 |
| **BulletGym Average** | 0.42 | 2.01 | 0.43 | 1.09 | **0.42** | **0.84** | 0.41 | 0.96 | 0.41 | 0.81 |

which the cost model is trained using binary labels only. The results indicate that PreSa outperforms these additional baselines. This performance gap is attributed to the baselines incorporating less information during learning, resulting in weaker performance.

## C.4 EVALUATION WITH FILTERED PREFERENCE DATASET

To investigate whether rewards matter when safety constraints are violated, we conducted experiments with PreSa using filtered preference data. Specifically, we excluded pairs where both segments were unsafe and pairs where the preferred segment was unsafe. The results, presented in Table 10, show that with such filtered preference data, the performance of our approach drops significantly and fails to learn safe behaviors.

## C.5 ABLATION STUDY

We explore the influence of several key hyperparameters used in PreSa. The results are shown in Tables 11, 12, 13, 14. These results indicate that while PreSa generally performs well across various hyperparameter settings, further fine-tuning can lead to improved performance.

## C.6 LEARNING CURVES

We train PreSa with the parameters in Table 4 for all 29 tasks. The learning curves are shown in Figure 6, 7, 8, and 9. In each figure, the dotted vertical line marks the point where $\pi_{\text{ref}}$ pretraining stops, while the dotted horizontal line indicates the cost threshold of 1.

Table 7: Evaluation results with imperfect binary safety labels: The noise level represents the percentage (e.g., 0.1 means 10%) of binary safety labels that are randomly selected for label flipping.

| Task | Noise Level | Binary Alignment | | BC-Safe-Seg | | Safe-RLHF (CDT) | | PreSa (Ours) | |
|------|-------------|---------|-------|---------|-------|---------|-------|---------|-------|
| | | reward↑ | cost↓ | reward↑ | cost↓ | reward↑ | cost↓ | reward↑ | cost↓ |
| BallRun | 0.0 | 0.31 | 4.79 | 0.37 | 1.13 | 0.35 | 1.65 | **0.19** | **0.09** |
| | 0.1 | **0.2** | **0.04** | 0.8 | 3.98 | 0.46 | 3.15 | 0.21 | 0.37 |
| | 0.2 | **0.2** | **0.03** | 0.71 | 3.51 | 0.52 | 2.87 | 0.24 | 0.92 |
| | 0.3 | 0.2 | 0.13 | 0.64 | 3.34 | 0.39 | 2.43 | 0.29 | 1.49 |
| BallCircle | 0.0 | **0.06** | **0.24** | 0.39 | 0.68 | 0.68 | 1.2 | 0.22 | 0.03 |
| | 0.1 | **0.14** | **0.03** | 0.48 | 1.57 | 0.66 | 1.34 | 0.22 | 0.07 |
| | 0.2 | **0.14** | **0.03** | 0.6 | 2.07 | 0.69 | 1.96 | 0.27 | 0.26 |
| | 0.3 | **0.16** | **0.07** | 0.53 | 1.52 | 0.68 | 2.21 | 0.32 | 0.49 |
| DroneRun | 0.0 | **0.11** | **0.17** | 0.17 | 5.97 | 0.47 | 3.12 | 0.16 | 0.33 |
| | 0.1 | **0.24** | **0.32** | 0.4 | 0.6 | 0.6 | 4.0 | 0.15 | 0.37 |
| | 0.2 | 0.25 | 0.31 | 0.22 | 0.82 | 0.36 | 2.07 | 0.14 | 0.44 |
| | 0.3 | 0.26 | 0.32 | 0.59 | 1.67 | **0.26** | **0.76** | 0.14 | 0.26 |
| DroneCircle | 0.0 | -0.23 | 1.59 | 0.61 | 1.9 | 0.61 | 0.87 | 0.54 | 0.72 |
| | 0.1 | 0.51 | 1.33 | 0.53 | 1.61 | 0.59 | 1.21 | 0.57 | 0.92 |
| | 0.2 | 0.53 | 1.37 | 0.71 | 2.79 | 0.58 | 1.44 | 0.59 | 1.2 |
| | 0.3 | 0.54 | 1.49 | 0.72 | 2.78 | 0.59 | 1.4 | 0.61 | 1.41 |

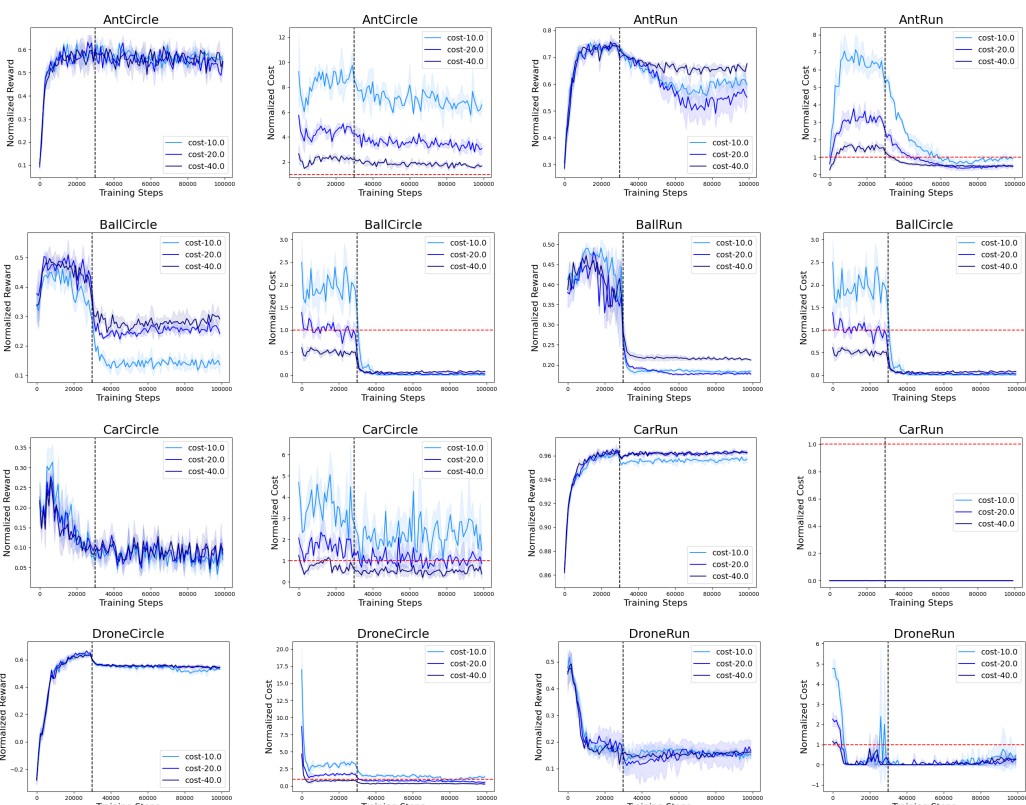

Figure 6: Training curves for the 8 tasks in BulletGym.

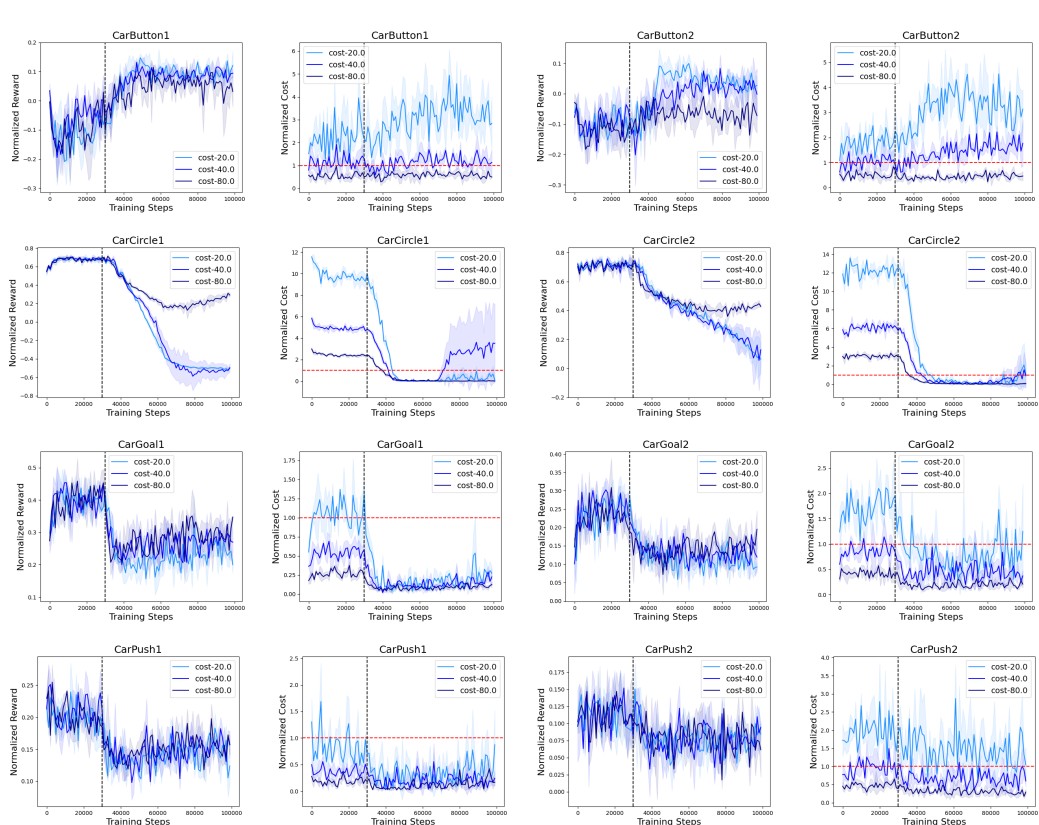

Figure 7: Learning curves for the 8 Car tasks in SafetyGym.

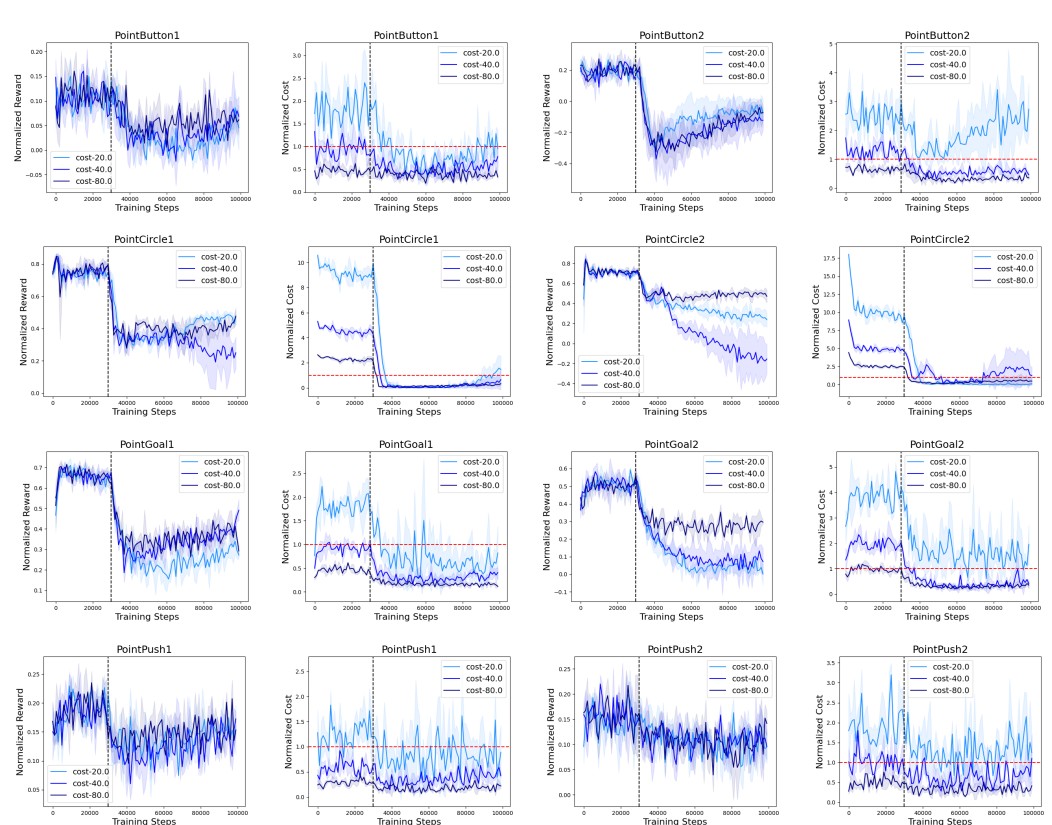

Figure 8: Learning curves for the 8 Point tasks in SafetyGym.

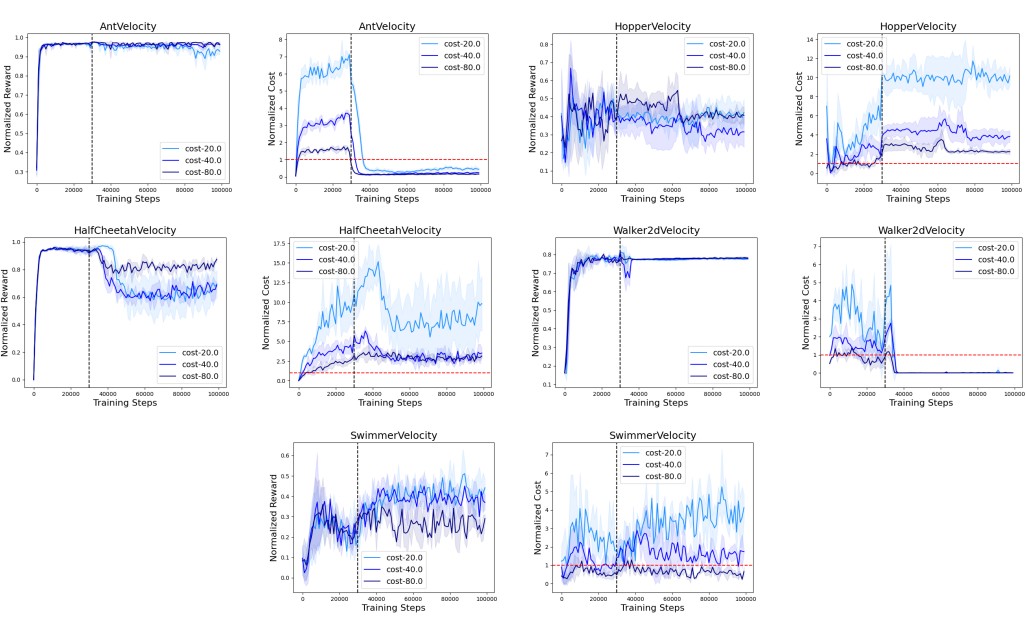

Figure 9: Learning curves for the 5 velocity constraint tasks in SafetyGym.

Table 8: Additional evaluation results of normalized reward and cost. The ↑ symbol indicates that the higher reward, the better, while the ↓ symbol signifies that the lower cost (up to a threshold of 1), the better. **Bold**: Safe agents whose normalized cost is below 1. **Orange**: Safe agent with the highest reward among approaches learning from offline synthetic human feedback.

| Task | Unified Comparison | | Binary Alignment | | BC-Safe-Seg | | Safe-RLHF (CDT) | | PreSa (Ours) | |
|---|---|---|---|---|---|---|---|---|---|---|
| | reward↑ | cost↓ | reward↑ | cost↓ | reward↑ | cost↓ | reward↑ | cost↓ | reward↑ | |
| BallRun | 0.27 | 0.77 | 0.31 | 4.79 | 0.37 | 1.13 | 0.35 | 1.65 | **0.19** | **0.09** |
| CarRun | 0.97 | 0.0 | **0.94** | **0.0** | 0.97 | 0.95 | 0.87 | 1.16 | 0.96 | 0.0 |
| DroneRun | 0.54 | 3.01 | **0.11** | **0.17** | 0.17 | 5.97 | 0.47 | 3.12 | 0.16 | 0.33 |
| AntRun | **0.48** | **0.4** | **0.09** | **0.01** | 0.66 | 1.38 | 0.72 | 1.04 | 0.61 | 0.63 |
| BallCircle | 0.44 | 0.21 | **0.06** | **0.24** | **0.39** | **0.68** | 0.68 | 1.2 | 0.22 | 0.03 |
| CarCircle | 0.54 | 1.47 | **0.06** | **0.35** | 0.56 | 1.76 | 0.57 | 0.84 | 0.08 | 0.91 |
| DroneCircle | 0.44 | 1.42 | -0.23 | 1.59 | 0.61 | 1.9 | 0.61 | 0.87 | 0.54 | 0.72 |
| AntCircle | 0.5 | 3.91 | 0.48 | 3.03 | 0.54 | 3.15 | 0.45 | 2.04 | 0.55 | 3.78 |
| **BulletGym Average** | 0.52 | 1.4 | 0.23 | 1.27 | 0.53 | 2.11 | 0.59 | 1.49 | 0.41 | 0.81 |

Table 9: Evaluation results for additional baselines using cost models trained exclusively on binary labels for Safe-RLHF.

| Task | Comparison Alignment | | Binary Alignment | | BC-Safe-Seg | | Safe-RLHF (CDT) | | Safe-RLHF (CDT) (Cost: binary label only) | PreSa (Ours) | |
|---|---|---|---|---|---|---|---|---|---|---|---|
| | reward↑ | cost↓ | reward↑ | cost↓ | reward↑ | cost↓ | reward↑ | cost↓ | reward↑ | cost↓ | reward↑ | cost↓ |
| BallRun | 0.27 | 0.77 | 0.31 | 4.79 | 0.37 | 1.13 | 0.35 | 1.65 | 0.31 | 1.52 | **0.19** | **0.09** |
| DroneCircle | 0.44 | 1.42 | -0.23 | 1.59 | 0.61 | 1.9 | 0.61 | 0.87 | 0.61 | 1.19 | **0.54** | **0.72** |

Table 10: Evaluation results using a filtered preference dataset, excluding pairs where both segments are unsafe or where the preferred segment is unsafe.

| Task | Preference w/ safety | | PreSa | |
|---|---|---|---|---|
| | reward↑ | cost↓ | reward↑ | cost↓ |
| BallRun | -0.08 | 5.46 | **0.19** | **0.09** |
| CarRun | **0.44** | **0.0** | **0.96** | **0.0** |
| DroneRun | **0.02** | **0.0** | **0.16** | **0.33** |
| AntRun | **0.35** | **0.46** | **0.61** | **0.63** |
| BallCircle | -0.0 | 5.03 | **0.22** | **0.03** |
| CarCircle | -0.1 | 3.23 | **0.08** | **0.91** |
| DroneCircle | **-0.26** | **0.03** | **0.54** | **0.72** |
| AntCircle | **0.0** | **0.0** | 0.55 | 3.78 |
| Average | 0.05 | 1.77 | **0.41** | **0.81** |

Table 11: Performance of PreSa with varying $\alpha$.

| Hyperparameter $\alpha$ | $\alpha = 0.2$ | | $\alpha = 0.4$ | | $\alpha = 0.6$ | | $\alpha = 0.8$ | |
|---|---|---|---|---|---|---|---|---|
| | reward↑ | cost↓ | reward↑ | cost↓ | reward↑ | cost↓ | reward↑ | cost↓ |
| BallRun | **0.19** | **0.09** | **0.19** | **0.1** | 0.38 | 2.8 | **0.19** | **0.12** |
| CarRun | **0.96** | **0.0** | **0.95** | **0.0** | **0.95** | **0.0** | **0.95** | **0.0** |
| DroneRun | **0.16** | **0.33** | 0.2 | 2.49 | 0.41 | 2.67 | 0.2 | 2.47 |
| AntRun | **0.61** | **0.63** | 0.65 | 1.85 | 0.63 | 2.88 | 0.68 | 1.86 |
| BallCircle | **0.22** | **0.03** | **0.15** | **0.03** | 0.3 | 1.06 | **0.17** | **0.04** |
| CarCircle | **0.08** | **0.91** | 0.2 | **0.87** | 0.32 | 1.01 | **0.23** | **0.99** |
| DroneCircle | **0.54** | **0.72** | **0.5** | **0.92** | 0.27 | 1.9 | 0.54 | 1.09 |
| AntCircle | 0.55 | 3.78 | 0.57 | 3.85 | 0.57 | 4.65 | 0.58 | 4.25 |
| **BulletGym Average** | **0.41** | **0.81** | 0.43 | 1.26 | 0.48 | 2.12 | 0.44 | 1.35 |

Table 12: Performance of PreSa with varying $\beta$.

| Hyperparameter $\beta$ | $\beta = 0.25$ | | $\beta = 0.5$ | | $\beta = 0.75$ | | $\beta = 1.0$ | |
|---|---|---|---|---|---|---|---|---|
| | reward↑ | cost↓ | reward↑ | cost↓ | reward↑ | cost↓ | reward↑ | cost↓ |
| BallRun | **0.18** | **0.11** | **0.18** | **0.09** | 0.34 | 2.67 | **0.19** | **0.09** |
| CarRun | **0.95** | **0.0** | **0.95** | **0.0** | **0.95** | **0.0** | **0.96** | **0.0** |
| DroneRun | 0.22 | 2.61 | 0.22 | 2.49 | 0.26 | 2.13 | **0.16** | **0.33** |
| AntRun | **0.6** | **0.88** | 0.64 | 1.37 | 0.53 | 2.14 | **0.61** | **0.63** |
| BallCircle | **0.16** | **0.03** | **0.16** | **0.03** | 0.3 | 1.09 | **0.22** | **0.03** |
| CarCircle | **0.22** | **0.83** | **0.2** | **0.79** | 0.27 | 1.24 | **0.08** | **0.91** |
| DroneCircle | 0.52 | 1.04 | **0.49** | **0.98** | 0.32 | 1.45 | **0.54** | **0.72** |
| AntCircle | 0.58 | 4.01 | 0.54 | 4.07 | 0.51 | 4.36 | 0.55 | 3.78 |
| **BulletGym Average** | 0.43 | 1.19 | 0.43 | 1.23 | 0.44 | 1.88 | **0.41** | **0.81** |

Table 13: Performance of PreSa with varying $\eta$.

| Hyperparameter $\eta$ | $\eta = 0.1$ | | $\eta = 0.5$ | | $\eta = 1.0$ | | $\eta = 5.0$ | |
|---|---|---|---|---|---|---|---|---|
| | reward↑ | cost↓ | reward↑ | cost↓ | reward↑ | cost↓ | reward↑ | cost↓ |
| BallRun | **0.19** | **0.09** | 0.66 | 4.4 | -0.53 | 4.68 | -0.17 | 5.46 |
| CarRun | **0.96** | **0.0** | **0.94** | **0.0** | **0.91** | **0.0** | **0.6** | **0.01** |
| DroneRun | **0.16** | **0.33** | 0.35 | 3.41 | 0.32 | 3.14 | 0.21 | 1.85 |
| AntRun | **0.61** | **0.63** | **0.6** | **0.7** | **0.57** | **0.71** | **0.5** | **0.65** |
| BallCircle | **0.22** | **0.03** | **0.13** | **0.14** | 0.05 | 3.14 | 0.09 | 5.45 |
| CarCircle | **0.08** | **0.91** | **0.11** | **0.46** | **0.07** | **0.42** | 0.01 | 9.2 |
| DroneCircle | **0.54** | **0.72** | **0.38** | **0.6** | 0.25 | 1.12 | **-0.26** | **0.04** |
| AntCircle | 0.55 | 3.78 | 0.51 | 3.51 | 0.48 | 6.17 | **0.0** | **0.0** |
| **BulletGym Average** | **0.41** | **0.81** | 0.46 | 1.65 | 0.26 | 2.42 | 0.12 | 2.83 |

Table 14: Ablation study of varying values of $\delta$.

| Task | $\delta = 0.65$ | | $\delta = 0.75$ | | $\delta = 0.85$ | | $\delta = 0.95$ | |
|---|---|---|---|---|---|---|---|---|
| | reward↑ | cost↓ | reward↑ | cost↓ | reward↑ | cost↓ | reward↑ | cost↓ |
| BallRun | 0.2 | 0.16 | 0.19 | 0.12 | 0.19 | 0.12 | 0.19 | 0.09 |
| DroneCircle | 0.57 | 0.94 | 0.56 | 0.87 | 0.56 | 0.88 | 0.54 | 0.72 |

