# OpenReview forum: "Offline Safe Policy Optimization From Human Feedback"
_ICLR.cc/2025/Conference — ICLR 2025 Conference Withdrawn Submission_

### Official Review · Reviewer_qgaE · 2024-10-21

**Soundness:** 2
**Presentation:** 2
**Contribution:** 2
**Rating:** 3
**Confidence:** 3

**Summary:**

This paper presents PreSa, for optimizing safe policies in offline reinforcement learning using human feedback. PreSa combines pairwise preference learning and safety alignment via binary safety labels, bypassing the need for explicit reward and cost models. The policy is optimized using a Lagrangian method within a constrained optimization framework. Empirical evaluations on synthetic human feedback demonstrate that PreSa outperforms state-of-the-art baselines in achieving high rewards while adhering to safety constraints across multiple tasks.

**Strengths:**

PreSa introduces a principled integration of preference and safety alignment, offering a novel method for offline safe RL. The technique eliminates the need for explicit reward and cost model learning, addressing the challenges of model inaccuracies.

**Weaknesses:**

1. Although PreSa avoids model inaccuracies without inferring the reward and cost model, uncertainty in offline datasets also leads to misspecifications. It is unfair to compare the two approaches without considering inaccuracies in your approach. I recommend the authors discuss specific types of uncertainties or misspecifications that might arise in their approach and how these compare to the inaccuracies in reward and cost model learning. This would provide a more balanced comparison.
2. Why do pairwise preferences of trajectory segments only consider rewards when costs exist in a CMDP environment?  I notice the dataset is split into ‘safe’ and ‘unsafe’ subsets, but do rewards really matter when a constraint has already been violated in the unsafe subset?
3. Such binary safety labels only consider hard constraint scenarios (correct me if I am wrong). Do the authors consider soft constraint scenarios where visiting an ‘unsafe’ trajectory segment after a sufficiently long number of steps would actually be safe, as indicated by the discounted cumulative version of constraint in Eq. (1)? I invite the authors to present a discussion on how PreSa might be adapted to handle soft constraints or discounted cumulative constraints, and what challenges this might present.
4. The experiments mainly include SafetyGym and BulletGym. I invite the authors to validate their results on more complicated environments, like CommonRoad, which offers a highly realistic simulation environment for developing and testing policies for autonomous systems in practical, safety-critical environments.

**Questions:**

1. It would be highly beneficial if the authors could also discuss the strategy for tuning the Lagrange multipliers used in Eq. 14. Since Eq. 14 employs a Lagrangian objective, tuning the updates of the Lagrange multipliers (such as the learning rate or initial values) poses significant challenges. Providing guidance on how to tune these parameters effectively would offer valuable insights to the community. I invite the authors to provide specific guidelines or heuristics they used for tuning the Lagrange multipliers or to discuss any challenges they encountered in this process.
2. Further, the Lagrangian methods do not necessarily guarantee safety, and it is challenging to recognize whether the optimum is reached. I invite the authors to comment on this point. I invite the authors to discuss potential strategies for addressing this limitation, such as incorporating additional safety checks or exploring alternative optimization techniques.
3. In Equation 14, it seems that some assumptions (e.g., Salter's condition) should be made to guarantee the existence of the strong duality. Do the authors miss this point? I invite the authors to explicitly state and justify the assumptions underlying their use of the Lagrangian method and to discuss how these assumptions might impact the applicability of their approach in different scenarios.

---

> ### Author Response · Authors · 2024-11-25
>
> We sincerely appreciate the time and effort you have dedicated to evaluating our work and providing insightful feedback. Below, we address the comments you raised and hope our responses help resolve the concerns.
>
> > **[W1]** Our approach directly learns the policy without requiring any intermediate model learning. We conduct additional experiments with imperfect, noisy safety feedback, and the results demonstrate the effectiveness of our approach across different levels of imperfection. For more details, please refer to General Response 2.
> ---
>
> > **[W2]** Rewards reflect task performance, measuring how efficiently the task is completed, while costs pertain to safety constraints, indicating how safely the agent behaves. We acknowledge that human pairwise preferences may be influenced by both rewards and costs when providing feedback. However, we assume that how humans provide preference feedback and whether this is influenced by costs largely depends on the instructions given to them prior to feedback collection. These instructions can be deliberately designed to elicit preferences based solely on task performance, regardless of safety considerations. A carefully crafted instruction for feedback collection can, to a large extent, guide humans to focus on rewards and minimize the influence of costs. To further investigate whether rewards matter when safety constraints are violated, we conducted experiments with PreSa using filtered preference data. Specifically, we excluded pairs where both segments were unsafe and pairs where the preferred segment was unsafe. The results, presented in Table 10 in the appendix of our revised manuscript, show that with such filtered preference data, the performance of our approach drops significantly and fails to learn safe behaviors.
> ---
>
> > **[W3]** To clarify, our approach is designed for scenarios with soft constraints. In the constraint term of Eq. (12), the parameter $\delta$ controls how strictly the policy adheres to the safety constraints. The approach becomes a hard-constrained problem only when $\delta = 1$.
> ---
>
> > **[W4]** Thank you for the suggestion. We evaluate the performance of our approach using the widely adopted DSRL benchmark, which provides pre-collected, well-formatted offline datasets for challenging tasks, making it particularly suitable for offline safe RL research. DSRL also offers standard evaluation protocols that ensure comprehensive and fair comparisons with baselines. Regarding the new domain, CommonRoad, we explored its feasibility for evaluation and found that additional work is needed, such as preparing offline datasets, generating synthetic feedback for policy learning, and designing cost functions for policy evaluation. Due to these requirements, we were unable to complete the necessary setup and provide results within the limited rebuttal period. However, we plan to conduct extended experiments with it and include the results in the future revision.
> ---
>
> > **[Q1]** To tune the Lagrange multiplier, we initialize its value at 0 and set the learning rate to start at 0.0001. We then use a binary search approach to gradually refine and narrow the range to identify the best learning rate. Ultimately, we determine two unified learning rates, one for BulletGym tasks and another for SafetyGym tasks.
> ---
>
> > **[Q2]** Thank you for the insightful comment highlighting a key limitation of Lagrangian methods. One way to address this is by applying constraint checks during the learning process to examine safety compliance. Alternatively, we might use offline data to pretrain the policy within feasible regions before transitioning to downstream tasks, ensuring better constraint satisfaction from the start. However, the effectiveness of these potential solutions would require further investigation, particularly in the offline learning setting with limited data. It is worth noting that these limitations are inherent to Lagrangian optimization itself and do not undermine the core contributions of our approach, as emphasized in General Response 1.
> ---
>
> > **[Q3]** Weak duality holds for our approach instead of strong duality, as a strictly feasible point may not always exist. We acknowledge that the absence of strong duality could limit the effectiveness of Lagrangian methods, potentially causing them to converge to suboptimal solutions. However, this limitation does not affect the overall effectiveness of our method for its intended purpose, as the weak duality property is sufficient to ensure valid bounds during optimization.

---

> > ### Comment · Reviewer_qgaE · 2024-11-25
> >
> > Thank you for the detailed response. I strongly encourage the authors to thoroughly address the concerns raised by the reviewers to refine the paper for a future conference submission. I shall keep my score.

---

### Official Review · Reviewer_4EHn · 2024-10-31

**Soundness:** 2
**Presentation:** 1
**Contribution:** 2
**Rating:** 3
**Confidence:** 3

**Summary:**

This paper introduces a novel framework called Offline Safe Policy Optimization from Human Feedback (POHF), which studies the RLHF problem under the setting of offline safe RL. The entire framework can also be viewed as an adaptation of the safe RLHF concept from LLMs to continuous control in RL. The whole framework directly optimizes the policy using Lagrangian objective contained reward and cost constraint, without the need for explicitly learning the reward and cost functions. By constructing a dataset based on DSRL, the paper demonstrates the effectiveness of its approach across multiple tasks and compares it with existing baseline methods, achieving significant improvements.

**Strengths:**

1. Optimizing Policies Without Explicit Reward and Cost Models: The method proposed in this paper enable policy optimization solely based on human preferences and safety feedback, eliminating the need for traditional reward and cost models. This significantly simplifies the policy optimization process, reducing inaccuracies associated with learning reward and cost models.

2. Comprehensive Experimental Validation: This paper conducts extensive experiments across 29 continuous control tasks, encompassing diverse environments and challenges. The results demonstrate that the proposed method outperforms existing baselines in multiple tasks, confirming its effectiveness and robustness.

**Weaknesses:**

1. The overall writing logic of the paper is relatively disjointed, which hinders comprehension, and there is considerable room for improvement. For instance, the connection between the challenges presented and the corresponding solutions in the abstract is quite weak. It is recommended that the background section 5.1.1 be integrated with the preliminaries in section 4 to enhance the paper's coherence.

2. The final method proposed in this paper is a combination of other approaches (CPL, HALOs, and the Lagrange objective in safe RL) within the POHF framework. And the POHF setting appears to be a transfer of Safe RL from LLMs to continuous control. Overall, the paper features a considerable amount of incremental work.

**Questions:**

1. The citation format for CPL in line 204 does not follow the correct guidelines. Incorrect Citation Format for Ethayarajh2024 in Line 260. Missing Equation Number for Formula in Line 298. Incorrect Citation Format for DSRL in Line 737. Incorrect Table Numbering Order. Lagrange should be written as a unified English name in the paper.

2. Why were experiments conducted on continuous control tasks, and is this method applicable and effective in discrete environments?
 Please discuss the potential challenges or modifications required to apply this method in discrete environments, or explain the specific focus on continuous control tasks.

3. Given the algorithm's moderate performance and robustness, have potential improvements been considered?

4.  Are there particular aspects of the method that could benefit from theoretical analysis or guarantees?

---

> ### Author Response · Authors · 2024-11-25
>
> We appreciate you dedicating your time and expertise to reviewing our work and providing thoughtful feedback. Below, we provide detailed responses to address your comments and questions.
>
> > **[W1]** Thanks for the suggestion. We have revised the manuscript to improve the paper flow for better readability.
> ---
>
> > **[W2]** Please see the General Response 1.
> ---
>
> > **[Q1]** Thank you for pointing out these issues. We have corrected them in the revised manuscript.
> ---
>
> > **[Q2]** To evaluate our approach, we adapt the widely used offline safe RL benchmark DSRL by generating synthetic human feedback, which includes challenging continuous control tasks from SafetyGym and BulletGym. They are well-recognized benchmarks in offline safe RL research. Additionally, our approach can be applied to discrete environments. Since our method directly learns the policy without requiring reward or cost learning and does not involve RL, it follows a supervised learning paradigm using an offline dataset. By simply modifying the neural network architecture of the policy network (e.g., adjusting the final layer to output categorical discrete actions), our method can be adapted to tasks in discrete environments.
> ---
>
> > **[Q3]** We respectfully acknowledge the reviewer's comment but have a different view on the evaluation of our performance. Our results outperform offline safe RL methods that use ground truth rewards and costs. Additionally, we significantly surpass all POHF baselines for both SafetyGym tasks and BulletGym tasks, as shown in the comprehensive results in Table 1. For future improvements, we aim to develop a novel approach that further eliminates Lagrangian optimization, simplifying the learning process.
> ---
>
> > **[Q4]** According to the theoretical convergence guarantee analysis of CPL [1], suppose that the optimal Lagrange multiplier is given and an unlimited number of preferences and binary labels are generated from a noisy rational regret-preference model using the expert advantage function $A^*$ which corresponds to a linear combination of reward and cost weighted by the Lagrange multiplier, PreSa can recover the optimal policy $\pi^\prime$.
> ---
>
> [1] Hejna, Joey, et al. "Contrastive Preference Learning: Learning from Human Feedback without Reinforcement Learning." The Twelfth International Conference on Learning Representations. 2024

---

### Official Review · Reviewer_sqxT · 2024-10-31

**Soundness:** 3
**Presentation:** 3
**Contribution:** 3
**Rating:** 5
**Confidence:** 4

**Summary:**

This paper introduces a framework for offline safe preference-based reinforcement learning. It assumes access to safe preferences in the form of binary labels instead of comparisons between two trajectories. Additionally, it proposes conducting preference learning through contrastive preference learning (CPL) and achieving safe alignment based on human-aware losses (HALOs). Safety alignment is then transformed into a constraint on the feasible policy set and thus integrated into preference learning. Finally, the paper presents a series of experiments on DSRL to evaluate the effectiveness of the proposed method.

**Strengths:**

- The paper is well-motivated. Circumventing explicit reward and cost learning could streamline the process and mitigate potential issues arising from explicit reward learning. Representing safety via binary safe labels instead of trajectory comparisons seems like a more reasonable and practical setting.
- Employing HALO-based safety alignment is intuitive and well-designed.
- The empirical evaluation and ablation studies are comprehensive, carefully demonstrating the effectiveness of the proposed approach.
- The paper is well-written and well-organized, with implementation details that are easy to understand.

**Weaknesses:**

- In Section 6.1, the paper generates labels based on cumulative rewards rather than estimated advantages. However, the original CPL paper shows the equivalence of advantage and optimal policy, based on which it uses an advantage that considers both reward and policy entropy to label the data. So I would like more justification as to why using cumulative rewards works here.
- To generate binary safety labels for trajectory segments, the paper uses reshaped thresholds based on segment length. It would be better to briefly discuss the influence of this approximation on safety performance.
- I notice that the $\delta$ in Eq. (11) is set to 0.9 or 0.95 instead of 1. Could you briefly explain the criteria for this setting and its influence on performance?
- In Eq.(4), does the symbol $P_{\pi_\theta}$ mean substituting $\pi^*$ in Eq.(3) with $\pi_\theta$? I think it would be helpful to explain this explicitly.
- I agree that circumventing explicit reward or cost learning is meaningful and brings benefits. Yet, the paper mentions that inaccuracies during reward and cost learning could be significant for performance. I would like to know if there is any supportive related work or experimental evidence to help me better understand this claim.

**Questions:**

See Weaknesses.

---

> ### Author Response · Authors · 2024-11-25
>
> We greatly appreciate the time and effort you have dedicated to reviewing our work and providing insightful feedback. Your feedback have been invaluable in enhancing the clarity and presentation of our contributions. Below, we provide detailed responses to address your comments.
>
> > **[W1]** Cumulative rewards are effective for generating synthetic feedback because optimal advantages are essentially a reshaped version of the reward, leading to the same optimal policy, particularly in domains with dense rewards. This justification is provided by the CPL authors in their responses to reviews of their ICLR 2024 submission. Thank you for pointing this out.
> ---
>
> > **[W2]** By reshaping the thresholds, we implicitly apply stricter constraints to each trajectory segment. The original cost threshold for the entire trajectory is proportionally distributed across the segments based on their lengths. Therefore, if we can learn a policy that satisfies the reshaped threshold for each segment, it will also satisfy the original threshold for the entire trajectory.
> ---
>
> > **[W3]** $\delta$ is a hyperparameter that controls the level of safety satisfaction in learning the safe policy. We conduct an ablation study on different values of $\delta$, and the results are presented in Table 14 in the appendix of our revised manuscript. From the results, we observe that the performance of PreSa remains generally stable and is relatively insensitive to the choice of $\delta$. However, as $\delta$ increases, the normalized cost of the learned policy slightly decreases, indicating that stricter cost constraints are being applied.
> ---
>
> > **[W4]** Thank you for pointing this out. As you observed, $\pi^\prime$ in Eq. (3) will be replaced with $\pi_\theta$, as $\pi^\prime$ is approximated by learning $\pi_\theta$ using Eq. (4). We make the necessary modifications in the revised manuscript.
> ---
>
> > **[W5]** Inaccuracies in reward and cost models can lead to either overestimation or underestimation of Q values, which can, in turn, impact policy learning in both RL and Safe RL. This issue is discussed in prior benchmarking work on offline safe RL [1].
> ---
>
> [1] Liu, Zuxin, et al. "Datasets and benchmarks for offline safe reinforcement learning." arXiv preprint arXiv:2306.09303 (2023).

---

> > ### Comment · Reviewer_sqxT · 2024-11-26
> >
> > Thank you for the response. However, some of my confusions remain unresolved. I agree that optimal advantages are a reshaped version of the reward, and if we treat it as the reward function and maximize it, we will obtain the same optimal policy. However, it is unclear whether this can also lead to the optimal policy under the CPL framework. It seems that the CPL framework does not rely on reward maximization, but rather on the relationship between the advantage and the optimal policy. Could you please point me to the derivation or provide further clarification on cumulative reward labeling within the CPL framework?
> >
> > Regarding the inaccuracies in reward and cost learning, I would recommend that the authors provide an empirical analysis of the reward and cost obtained from Safe-RLHF, which would better support your claims.

---

### Official Review · Reviewer_e7Rv · 2024-10-31

**Soundness:** 2
**Presentation:** 3
**Contribution:** 2
**Rating:** 3
**Confidence:** 4

**Summary:**

The paper addresses the challenge of learning a safe policy from offline human feedback, which comprises both reward preferences and binary safety labels. To tackle this, it proposes a framework containing two parts: 1) maximizing rewards using traditional contrastive preference learning and 2) encouraging safety by guiding the policy toward generating safer trajectories with higher probability. These two methods are then integrated using a Lagrangian approach. Extensive experiments across diverse tasks demonstrate the effectiveness of this approach.

**Strengths:**

1. The paper is well-structured and easy to understand. The authors convey their settings, motivations, and methods clearly, presenting a logical flow.
2. The experiments on the simulation tasks are extensive, containing 29 tasks across two environments. The baseline methods include both conventional offline RL methods and PbRL methods, which is good.

**Weaknesses:**

1. The paper states that "it is relatively scarce and expensive to collect safety preferences." However, I believe that obtaining direct safety labels for each trajectory, especially perfectly accurate ones, is even more challenging and, in many cases, unrealistic.
2. This paper is more like incremental work, primarily combining some existing methods (e.g., contrastive preference learning for preference alignment, human-aware losses for safety concerns) to solve a new offline PbRL setting with reward preferences and safety binal labels. While the practical importance of this problem setting is acknowledged, the novelty of the approach could be seen as limited.
3. The offline dataset contains both reward preferences and safety labels, which the paper addresses separately and later integrates in a final optimization step. Specifically, the proposed method uses reward preferences to promote a reward-maximizing policy, while safety labels are employed to encourage safer behaviors within the dataset. However, exploring methods that leverage these two types of feedback in a more unified manner, rather than treating them independently, could potentially eliminate the need for constrained optimization (via the Lagrange method), thus simplifying the optimization process and reducing computational complexity.
4. Although the paper introduces many baseline methods, I think it is indeed unnecessary to compare so many offline RL methods with ground-truth reward and cost since the two settings are different. Instead, the authors should investigate more efficient offline RLHF methods with offline labels for fair comparison. Please see the following questions for details.

**Questions:**

1. Could the authors include additional offline RLHF methods for comparison? For instance, a unified preference label prioritizing safety could be designed, where safer trajectories (according to the safety label) and higher-reward trajectories (according to reward preference) are preferred. This unified preference could then be used for policy alignment through CPL or RLHF. Additionally, it may be feasible to train a cost model to predict the cost of state-action pairs based on the safety label, which could then be integrated into the RLHF phase.
2. How are the cost preferences for Safe-RLHF generated, given that the offline dataset contains only binary safety labels? I believe that each binary label per trajectory segment conveys more information than preference, and comparing with a different dataset is not fair.
3. Can the authors study the situation where the safety labels are not perfect?

---

> ### Author Response · Authors · 2024-11-25
>
> Thank you for your time and effort in reviewing our paper and providing valuable feedback. Some of your comments and questions are addressed in the General Response. Below, we provide detailed responses to the remaining points and hope that our clarifications address your concerns.
>
> > **[W1]** When providing feedback, humans evaluate each trajectory individually [1], assigning feedback based on their assessment. Binary label selection requires less effort compared to comparison-based feedback, which involves double evaluations for each pair of trajectories [1][2]. Furthermore, in the context of an MDP, safety evaluations are often tied to the cumulative cost of a trajectory, which reflects its level of safety, and a predefined cost threshold, which determines whether a trajectory is deemed safe. Binary labels are derived from both the cumulative cost and the threshold, whereas pairwise cost preferences, based solely on comparisons of cumulative costs, lack critical information about the threshold required for evaluating safety violations. Therefore, binary labels are both more low-effort and essential for safe policy learning.
> ---
>
> > **[W2]** Please see the General Response 1.
> ---
>
> > **[W3]** Thank you for the thoughtful comments. Eliminating the constrained optimization is also an important objective for us, and we aim to explore novel approaches to achieve this in future work.
> ---
>
> > **[W4]** Offline RL methods are included as oracles to provide an upper bound on performance, following the evaluation protocol of prior PbRL studies [3][4][5]. Similarly, those studies also utilize SAC in experiments with access to ground truth rewards, where SAC functions as an oracle, setting a benchmark for the maximum achievable performance.
> ---
>
> > **[Q1]** We conduct experiments on the suggested baselines and included the results in the appendix of our revised manuscript in Table 8 and 9. The results show that PreSa outperforms these additional baselines, as the baselines incorporate less information during learning, leading to weaker performance. Due to time constraints during the rebuttal period, we focus on tasks in BulletGym. We plan to perform more extensive experiments on tasks in SafetyGym and include those results in the main paper in a future revision.
> ---
>
> > **[Q2]** *For cost preferences*: We generate the cost preferences using ground truth cost values provided in the DSRL offline dataset.
> *For experimental comparison with Safe-RLHF*: Safe-RLHF is the SOTA method for policy learning in contextual bandits for LLMs, we strictly follow its experimental setup, including the use of cost preferences. We acknowledge that the comparison may not be fair, as Safe-RLHF learns from more information, while PreSa only learns from binary labels for safety alignment, making it a more challenging task. Our goal is to demonstrate that, even with less information, our approach can still outperform SOTA baselines, and the results reflect this.
> ---
>
> > **[Q3]** Please see the General Response 2.
> ---
>
> [1] Casper, Stephen, et al. "Open Problems and Fundamental Limitations of Reinforcement Learning from Human Feedback." Transactions on Machine Learning Research.
>
> [2] Ethayarajh, Kawin, et al. "Kto: Model alignment as prospect theoretic optimization." arXiv preprint arXiv:2402.01306 (2024).
>
> [3] Lee, Kimin, et al. "PEBBLE: Feedback-Efficient Interactive Reinforcement Learning via Relabeling Experience and Unsupervised Pre-training." International Conference on Machine Learning. PMLR, 2021.
>
> [4] Park, Jongjin, et al. "SURF: Semi-supervised Reward Learning with Data Augmentation for Feedback-efficient Preference-based Reinforcement Learning." International Conference on Learning Representations. 2022.
>
> [5] Liu, Runze, et al. "Meta-reward-net: Implicitly differentiable reward learning for preference-based reinforcement learning." Advances in Neural Information Processing Systems 35 (2022): 22270-22284.

---

> > ### Comment · Reviewer_e7Rv · 2024-11-25
> >
> > Thank you for the authors' response. However, I believe the paper still requires improvements, both in terms of its theoretical contributions and additional empirical evaluations. The responses provided do not fully address my concerns, and I will maintain my original score.

---

### Author Response · Authors · 2024-11-25
**General Response**

We sincerely appreciate the time and effort that all the reviewers have dedicated to reviewing our paper and providing thoughtful feedback. Below, we address questions and concerns raised by multiple reviewers, and we hope our responses help clarify these points.

1. **On novelty**: Although our work builds upon and is inspired by existing research, we would like to highlight the key contributions of our work as follows.
> * **Extending the HALO Framework**: The original HALO framework was developed for fine-tuning LLMs using a contextual bandit approach. We introduce a novel extension of this framework to RL, which corresponds to sequential decision-making, a significantly more challenging problem that is neither straightforward nor intuitive.
> * **Integrating Preference and Safety Alignment Modules**: Combining preference learning with safety learning is a non-trivial task. To address this, we reformulate the safety learning objective to define a feasible policy set that serves as the foundation for a unified framework where these two modules are naturally integrated. The unified objective also implicitly aligns with the standard objective of safe RL, maintaining consistency with established principles.
> * **Challenging Task in Safety Feedback**: Safety evaluation in RL typically focuses on the cumulative cost of trajectories, indicating the degree of safety, and a cost threshold that determines whether a policy is safe. Unlike prior work tailored for LLM tasks, which incorporates both pairwise cost comparisons and binary safety labels for cost learning, we simplify the setup by considering only binary labels, omitting pairwise cost comparisons. Despite this more challenging problem setting, our approach outperforms baseline methods, including the one that leverages both pairwise comparisons and binary labels.
---

2. **On the evaluation with imperfect safety feedback**: We conduct experiments by introducing noises to the safety feedback to simulate different levels of imperfection. For each level, we randomly select a subset of the safety feedback and flip its True/False labels. The results, presented in Table 7 in the appendix of our revised manuscript, show that PreSa outperforms the baselines across various levels of imperfect safety feedback, with performance gradually decreasing as the noise level increases.

---

### Note · Authors · 2024-12-22

I have read and agree with the venue's withdrawal policy on behalf of myself and my co-authors.